# Disturbances in Mitochondrial Network, Biogenesis, and Mitochondria-Mediated Inflammatory Responses in Selected Brain Structures of Rats Exposed to Lead (Pb) During Prenatal and Neonatal Development

**DOI:** 10.3390/ijms262411907

**Published:** 2025-12-10

**Authors:** Mikołaj Chlubek, Magdalena Gąssowska-Dobrowolska, Agnieszka Kolasa, Maciej Tarnowski, Patrycja Tomasiak, Agnieszka Maruszewska, Katarzyna Barczak, Irena Baranowska-Bosiacka

**Affiliations:** 1Department of Biochemistry and Medical Chemistry, Pomeranian Medical University, Powstańców Wlkp. 72, 70-111 Szczecin, Poland; mikolaj.chlubek@gmail.com; 2Department of Cellular Signalling, Mossakowski Medical Research Institute, Polish Academy of Sciences, 02-106 Warsaw, Poland; mgassowska@imdik.pan.pl; 3Department of Histology and Embryology, Pomeranian Medical University, Powstańców Wlkp. 72, 70-111 Szczecin, Poland; agnieszka.kolasa@pum.edu.pl; 4Department of Physiology in Health Sciences, Pomeranian Medical University, Żołnierska 54, 70-210 Szczecin, Poland; maciej.tarnowski@pum.edu.pl; 5Institute of Physical Culture Sciences, University of Szczecin, 70-240 Szczecin, Poland; patrycja.tomasiak@pum.edu.pl; 6Institute of Biology, University of Szczecin, Wąska 13, 71-415 Szczecin, Poland; agnieszka.maruszewska@usz.edu.pl; 7Department of Endodontic Surgery, Pomeranian Medical University, Powstańców Wlkp. 72, 70-111 Szczecin, Poland; katarzyna.barczak@pum.edu.pl

**Keywords:** lead neurotoxicity, mitochondrial biogenesis, mitochondrial fission/fusion, rat brain, developmental lead exposure, cGAS-STING pathway

## Abstract

Lead (Pb) disrupts mitochondrial function, but its impact on the mitochondrial dynamics and biogenesis during early brain development remains insufficiently understood. This study aimed to investigate the effects of pre- and neonatal Pb exposure on the processes involved in mitochondrial network formation in the brains of rat offspring, simulating environmental exposure. We quantified mRNA expression (qRT-PCR) and protein levels (ELISA) of key mitochondrial fusion (Mfn1, Mfn2, Opa1), fission (Drp1, Fis1) regulators, as well as biogenesis markers (PGC-1α, TFAM, NRF1) in the hippocampus, forebrain cortex, and cerebellum of rats exposed to Pb. Mitochondrial ultrastructure was evaluated using transmission electron microscopy (TEM), and the expression of mitochondrial electron transport chain (ETC) genes was analysed (qRT-PCR). Furthermore, to examine the involvement of the cGAS–STING pathway in Pb-induced neuroinflammation, we measured the expression of ISGs (qRT-PCR), TBK1 phosphorylation (Western blot), and 2′,3′-cGAMP synthesis (ELISA). Our results showed that Pb exposure markedly reduced PGC-1α and region-specific NRF1 levels, broadly supressed fusion proteins (Mfn1, Mfn2, Opa1), increased Fis1, and depleted Drp1. ETC gene expression (*mtNd1*, *mtCyb* and *mtCo1*) were upregulated in a brain-structure-dependent manner. These molecular changes were accompanied by pronounced mitochondrial morphological abnormalities. Despite upregulation of *Mx1*, *Ifi44*, and *Sting1*, along with synthesis of 2′3′-cGAMP, TBK1 activation was not detected. All these findings demonstrate that early-life Pb exposure, even low-dose, disrupts mitochondrial biogenesis and the fusion–fission machinery, thus impairs brain energy homeostasis, and implicates mitochondria as central mediators of Pb-induced neuroinflammation and neurodevelopmental toxicity.

## 1. Introduction

Lead (Pb) is a well-established neurotoxin, with the developing nervous system exhibiting particular vulnerability to its detrimental effects [1]. Numerous studies have demonstrated that both prenatal and early childhood exposure to Pb can result in persistent neurological impairments, including reduced cognitive performance, decreased IQ, learning disabilities, and behavioral disturbances [2,3,4,5,6].

Although the neurotoxic effects of Pb have been widely investigated, the molecular mechanisms that underlie these effects are not yet fully understood. Among the proposed mechanisms, disruption of mitochondrial function has emerged as a critical pathway in Pb-induced neurotoxicity. Growing evidence indicates that mitochondria are primary targets of Pb toxicity [7,8,9], with studies identifying them as major sites of Pb accumulation in neural tissue [10].

Mitochondria are central to numerous essential biochemical processes, including adenosine triphosphate (ATP) production, fatty acid biosynthesis, intracellular reactive oxygen species (ROS) generation, oxidative phosphorylation (OXPHOS), thermogenesis, and calcium (Ca^2+^) homeostasis [11]. In the nervous system, mitochondria are indispensable for neuronal energy metabolism, calcium signaling, and the regulation of apoptotic pathways, all of which are crucial for normal brain development and function. The maintenance of stable mitochondrial morphology is fundamental for sustaining neuronal metabolic activity [12].

Our previous studies using this model of Pb toxicity demonstrated that perinatal exposure to Pb disrupts mitochondrial function, resulting in a reduced inner mitochondrial membrane potential (ΔΨm), decreased concentrations of high-energy nucleotides (ATP), a diminished adenine nucleotide pool (TAN), and a lowered adenylate energy charge (AEC). These alterations were accompanied by inhibition of Na^+^/K^+^-ATPase activity, increased oxidative stress, as evidenced by elevated intracellular and mitochondrial ROS levels, and changes in mitochondrial morphology [13].

However, the impact of Pb exposure on mitochondrial biogenesis and dynamics—key processes responsible for maintaining mitochondrial integrity and quality control—remains poorly understood [14,15]. In healthy cells, mitochondria undergo continuous and tightly regulated cycles of fission, fusion, mitophagy, and intracellular transport, collectively referred to as mitochondrial dynamics. These processes determine mitochondrial morphology, number, quality, and subcellular distribution, thereby directly influencing overall mitochondrial function.

Maintaining a proper balance between these opposing mechanisms is essential for optimal mitochondrial and cellular function, including efficient energy production and regulation of key biological processes such as cell motility, differentiation, cell cycle progression, senescence, and apoptosis [16,17,18]. An imbalance in the processes of mitochondrial fission, fusion, mitophagy, and transport leads to mitochondrial dysfunction and has been implicated in the pathogenesis of a broad range of diverse diseases and pathologies associated with neurodegeneration and impaired neuronal function, including neurodegenerative and neurodevelopmental disorders [19,20,21,22,23,24,25].

Mitochondrial fission contributes to the redistribution of mitochondria within the cell and their equal segregation during mitosis, and plays a critical role in quality control through the selective elimination of damaged mitochondria via mitophagy. In contrast, fusion facilitates the mixing of mitochondrial contents, including mitochondrial DNA and proteins, enabling the repair of partially damaged organelles and supporting the maintenance of mitochondrial function [11,17,18].

Mitochondrial fission is primarily mediated by the cytosolic GTPase dynamin-related protein 1 (Drp1). Upon activation, Drp1 is recruited to the outer mitochondrial membrane by various adaptor proteins, including mitochondrial fission protein 1 (Fis1), mitochondrial fission factor (Mff), and other mitochondrial receptors such as MiD49 and MiD51 [26,27]. Drp1 interacts with these adaptors and oligomerizes into spiral-like structures that encircle the mitochondrion. Through GTP hydrolysis, Drp1 constricts these rings, leading to scission of the mitochondrial membrane and promoting mitochondrial fragmentation [28,29].

Conversely, mitochondrial fusion is tightly regulated by a set of GTPase proteins. Mitofusin-1 (Mfn1) and mitofusin-2 (Mfn2), both located in the outer mitochondrial membrane, mediate the tethering and subsequent fusion of adjacent mitochondria. Fusion of the inner mitochondrial membrane is orchestrated by optic atrophy protein 1 (Opa1), a dynamin-related GTPase localized to the inner membrane, which also contributes to the maintenance of cristae architecture and mitochondrial bioenergetics function [30].

Mitochondrial damage induced by Pb, as demonstrated in both our current and previous studies, may result in the release of mitochondrial components—normally confined within the organelle—into the cytosol and extracellular space [31,32]. These mitochondrial-derived molecules, including mitochondrial DNA (mtDNA), mitochondrial reactive oxygen species (mtROS), adenosine triphosphate (ATP), mitochondrial transcription factor A (TFAM), cardiolipin, and cytochrome c, are collectively referred to as danger signals or mitochondrial alarmins [31,33,34,35,36,37,38,39,40,41].

Functionally, these molecules act as damage-associated molecular patterns (DAMPs) capable of activating innate immune signalling pathways. Specifically, they engage pattern recognition receptors (PRRs) such as RIG-I-like receptors (RLRs), NOD-like receptors (NLRs), cyclic GMP-AMP synthase (cGAS), and Toll-like receptors (TLRs), which are located on the plasma membrane, within endosomes, or in the cytoplasm of immune and glial cells in the central nervous system (CNS) [42,43,44]. Activation of these receptors initiates pro-inflammatory signalling cascades that contribute to the onset and persistence of neuroinflammation [45,46,47].

Consequently, Pb-induced mitochondrial pathology may represent a critical upstream trigger linking environmental neurotoxicant exposure to chronic neuroinflammation and, ultimately, neurodegeneration. Elevated circulating levels of mtDAMPs, particularly mtDNA, can activate the cytosolic DNA sensor cyclic GMP-AMP synthase (cGAS) and its signalling effector–stimulator of interferon genes (STING), initiating the cGAS–STING pathway and leading to the production of interferon-stimulated genes (ISGs) and other inflammatory mediators [31,48]. Excessive activation of the cGAS–STING pathway has been implicated in various pathological conditions, including neurodegenerative disorders. Aberrant stimulation of this pathway contributes to chronic neuroinflammation and neuronal loss. In chronic neurodegeneration, sustained cGAS–STING signalling drives overproduction of type I interferons (IFN-I), which in turn promote microglial activation toward a proinflammatory phenotype. This maladaptive immune response exacerbates neuronal injury and accelerates disease progression [49].

Although Pb-induced mitochondrial dysfunction has been investigated in various models of both acute and chronic exposure, the specific mechanisms by which Pb disrupts mitochondrial dynamics—particularly the processes of fusion and fission—remain incompletely understood. Another relatively underexplored area is the effect of Pb exposure on mitochondrial biogenesis. Notably, it is still unclear whether exposure to low, environmentally relevant concentrations of Pb, resulting in blood lead levels below 10 μg/dL in offspring (a threshold previously considered “acceptable” or “safe” for humans), during the prenatal and early postnatal period, can adversely affect mitochondrial function. Moreover, no studies to date have examined the impact of perinatal Pb exposure on the cGAS–STING pathway.

Therefore, this study aimed to investigate the effects of pre- and neonatal Pb exposure on mitochondrial biogenesis and the processes involved in mitochondrial network formation in the brains of rat offspring. Specifically, we assessed the mRNA expression and protein levels of key regulators of mitochondrial fusion and fission, including mitofusin 1 (Mfn1), mitofusin 2 (Mfn2), mitochondrial dynamin-like GTPase (Opa1), dynamin-related protein 1 (Drp1), and mitochondrial fission 1 protein (Fis1), in the hippocampus, forebrain cortex, and cerebellum of rats exposed to Pb during the pre- and neonatal periods, simulating environmental exposure. In addition, we evaluated the mRNA expression and protein levels of mitochondrial biogenesis markers such as peroxisome proliferator-activated receptor γ coactivator 1α (PGC-1α), mitochondrial transcription factor A (TFAM), and nuclear respiratory factor 1 (NRF1). Mitochondrial ultrastructure in all examined brain regions was also assessed using transmission electron microscopy (TEM), and the expression of genes encoding components of the mitochondrial electron transport chain (ETC) was analysed. Furthermore, we examined the involvement of the cGAS–STING pathway in Pb-induced neuroinflammation and neuropathology.

## 2. Results

### 2.1. Perinatal Exposure to Pb Increased Pb Concentrations in Whole Blood and Brain

The detailed results of Pb concentrations in the brain and blood were previously published [50]. Briefly, Pb levels in whole blood were significantly elevated in the Pb-exposed group compared to controls (6.86 µg/dL vs. 0.93 µg/dL). In the analysed brain regions, Pb concentrations ranged from 7.10 to 7.48 µg/dL in the Pb group and from 0.04 to 0.26 µg/dL in controls.

### 2.2. Effects of Perinatal Pb Exposure on Mitochondrial Ultrastructure and Homeostasis

#### 2.2.1. Perinatal Pb Exposure Altered Mitochondrial Ultrastructure

Microscopic analysis revealed pathological ultrastructural changes in mitochondria across all examined brain regions in rats exposed to Pb during the pre- and neonatal period. The preserved ultrastructure of mitochondria in the control group (Figure 1(A)A,C,E and Figure 1(B)A’,C’,E’) confirms proper perfusion and fixation procedures, indicating that the changes observed in the Pb group result from Pb neurotoxicity. In neurons from Pb-exposed animals (Figure 1(A)B,D,F and Figure 1(B)B’,D’,F’), mitochondria exhibited structural abnormalities, appearing elongated, swollen, or shrunken.

#### 2.2.2. Perinatal Pb Exposure Affected Mitochondrial Dynamics in the Brain of Rat Offspring

The observed morphological differences in mitochondria between control and Pb-exposed animals may result, at least in part, from altered expression of proteins involved in mitochondrial fusion and fission. This hypothesis is supported by qRT-PCR and ELISA analysis of key regulatory proteins: *Mfn1*, *Mfn2*, and *Opa1* (fusion), and *Drp1* and *Fis1* (fission).

qRT-PCR analysis showed that Pb exposure significantly increased *Mfn1* mRNA expression in the forebrain cortex and cerebellum by approximately 27% (*p* = 0.0265) and 96% (*p* < 0.0001), respectively, and *Mfn2* mRNA expression in the hippocampus by approximately 38% (*p* = 0.0019), relative to controls (Figure 2(A.1,B.1)). However, these transcriptional changes were not reflected at the protein level. ELISA results revealed significant decreases in *Mfn1* and *Mfn2* concentrations by approximately 63% (*p* = 0.0004) and 45% (*p* = 0.0122) in the hippocampus and cerebellum, and reductions of approximately 63% (*p* < 0.0001), 33% (*p* = 0.0011), and 22% (*p* = 0.0001) in the hippocampus, forebrain cortex, and cerebellum, respectively (Figure 2(A.2,B.2)). A possible explanation for the inconsistency between mRNA and protein levels of mitochondrial fusion/fission proteins observed in our studies may result from the effect of Pb on post-transcriptional processes through translation inhibition [50], miRNA-mediated regulation [51], and altered mRNA stability/degradation [52]. These processes are prominent in mainly neuronal models, where Pb-induced ROS and Ca^2+^ dysregulation amplify post-transcriptional control, leading to attenuated protein responses during neurotoxicity [53,54].

Similarly, *Opa1* protein levels were significantly reduced in the hippocampus (by approximately 41%, *p* < 0.0001) and cerebellum (by nearly 34%, *p* = 0.0011), despite a 66% increase (*p* < 0.0001) in *Opa1* mRNA expression in the cerebellum (Figure 2(C.1,C.2)).

Furthermore, pre- and neonatal exposure to Pb induced significant alterations in the gene expression of mitochondrial fission proteins, including *Drp1* and *Fis1*. A significant decrease in *Drp1* mRNA expression (approximately 54%, *p* = 0.0115) was observed in the hippocampus, while a substantial increase (approximately 132%, *p* = 0.0009) was detected in the cerebellum (Figure 3(A.1)). In contrast, ELISA analysis revealed a significant decrease in Drp1 protein levels across all analyzed brain structures: by approximately 32% in the hippocampus, 48% in the forebrain cortex, and 32% in the cerebellum (*p* < 0.0001 for all) (Figure 3(A.2)).

Regarding *Fis1*, a significant increase in both mRNA (approximately 51%, *p* = 0.0016) and protein concentration (approximately 27%, *p* = 0.0006) was observed in the forebrain cortex (Figure 3(B.1,B.2)). In the cerebellum, *Fis1* protein concentration increased by approximately 46% (*p* = 0.0158), despite only a non-significant upward trend in mRNA expression (Figure 3(B.2)).

These findings indicate a disruption of mitochondrial fusion and fission dynamics in the brains of Pb-exposed offspring. The significant depletion of proteins involved in these processes suggests impaired mitochondrial dynamics and function, potentially contributing to reduced energy production, increased oxidative stress, and subsequent neuronal dysfunction.

#### 2.2.3. Perinatal Exposure to Pb Affected Mitochondrial Biogenesis in the Brain of Rat Offspring

Next, we assessed the impact of pre- and neonatal Pb exposure on mitochondrial biogenesis. Analysis of gene expression for key regulators of mitochondrial generation—*Pqargc1* (*PGC-1α*), *Tfam1* (*TFAM*), and *Nrf1* (*NRF1*)—revealed region-specific changes. *Pqargc1* mRNA expression was significantly increased only in the forebrain cortex by approximately 139% (*p* = 0.0221) compared to the control (Figure 4(A.1)). Conversely, *Tfam1* expression was significantly reduced in the cerebellum by approximately 75% (*p* = 0.0087) (Figure 4(B.1)). *Nrf1* mRNA levels were also markedly decreased in both the forebrain cortex and cerebellum by approximately 75% (*p* = 0.0250) and 90% (*p* = 0.0003), respectively, while remaining unchanged in the hippocampus (Figure 4(C.1)).

Despite the observed increase in *Pqargc1* mRNA, ELISA analysis showed a significant reduction in PGC-1α protein levels across all analyzed regions: by approximately 3% in the hippocampus (*p* = 0.00275), 8% in the forebrain cortex (*p* = 0.013), and 7.5% in the cerebellum (*p* = 0.0025) (Figure 4(A.2)). Similarly, NRF1 protein levels were significantly reduced by approximately 14% in the forebrain cortex (*p* = 0.0346) and 21% in the cerebellum (*p* = 0.0437) (Figure 4(C.2)). In contrast, TFAM protein levels remained unchanged in all examined brain structures (Figure 4(B.2)).

#### 2.2.4. Perinatal Pb Exposure Induced Overexpression of Mitochondrial ETC Complexes in the Brain of Rat Offspring

Disruptions in mitochondrial fusion and fission processes can impair oxidative phosphorylation. To further assess mitochondrial function, we examined the expression of genes encoding subunits of the mitochondrial electron transport chain (ETC) complexes. In the hippocampus, no significant differences in the mRNA expression of *mtNd1*, *mtSdha*, *mtCyb*, or *mtCo1* (subunits of complexes I, II, III, and IV, respectively) were observed between Pb-exposed and control rats (Figure 5A).

In contrast, in the forebrain cortex of Pb-exposed rats, we observed a significant increase (*p* = 0.0477) in *mtNd1* mRNA expression by approximately 104% compared to controls (Figure 5B). In the cerebellum, gene expression levels of *mtNd1*, *mtCyb*, and *mtCo1* were significantly elevated by approximately 317% (*p* = 0.0138), 117% (*p* < 0.0001), and 40% (*p* = 0.0114), respectively (Figure 5C).

These findings suggest region-specific upregulation of ETC complex subunits—particularly complexes I, III, and IV—in the forebrain cortex and cerebellum in response to perinatal Pb exposure, potentially as a compensatory mechanism to maintain mitochondrial function under toxic stress.

### 2.3. Effects of Perinatal Pb Exposure on the Expression of the Interferon-Stimulated Genes (ISGs) and Activation of cGAS in the Brain of Adult Rats

qRT-PCR analysis showed that perinatal Pb exposure significantly increased the expression of *Mx1*, *Ifi44*, and *Sting1* in the hippocampus by approximately 665% (*p* < 0.0001), 170% (*p* = 0.0001), and 45% (*p* = 0.0007), respectively (Figure 6A). In the forebrain cortex, mRNA levels of these genes were also significantly elevated: *Mx1* by 725% (*p* = 0.0183), *Ifi44* by 64% (*p* = 0.0164), and *Sting1* by 153% (*p* = 0.0154) (Figure 6B). In the cerebellum, a significant increase was observed only in *Mx1* expression (by 212%, *p* = 0.0317), while *Ifi44* levels remained unchanged, and *Sting1* expression was significantly decreased by 36% (*p* = 0.0043) (Figure 6C).

The observed upregulation of *Mx1*, *Ifi44*, and *Sting1* is indicative of type I interferon (IFN-I) pathway activation, often triggered by cellular stress, infection, or cytosolic self-derived mitochondrial DNA (mtDNA). To assess the potential role of the cGAS–STING pathway in this response, we measured the concentration of 2′,3′-cyclic GMP-AMP (2′,3′-cGAMP), the second messenger synthesised by cGAS upon cytosolic DNA binding, as well as activation of the TBK1, measured by the levels and phosphorylation status of TBK1 at Ser172.

ELISA analysis revealed a significant increase in 2′,3′-cGAMP levels across all brain regions of Pb-exposed rats: by 140% in the hippocampus (*p* = 0.0006), 35% in the forebrain cortex (*p* = 0.0145), and 135% in the cerebellum (*p* = 0.009) (Figure 7A–C), confirming activation of the cGAS following perinatal Pb exposure.

However, Western blot analysis did not show significant changes in TBK1 activation, as neither phosphorylation at Ser172 nor total TBK1 protein levels differed significantly in most brain regions (Figure 8A,B). The only exception was a significant reduction in total TBK1 protein levels in the hippocampus (*p* < 0.05) (Figure 8B,C).

These results suggest that perinatal Pb exposure activates cGAS and induces ISG expression in a TBK1-independent manner.

## 3. Discussion

Our findings demonstrate that even low-dose pre- and neonatal Pb exposure can disrupt both mitochondrial biogenesis and the dynamic processes of fusion and fission involved in mitochondrial network formation in a brain-structure-specific manner. These alterations impair brain energy metabolism and may contribute to neurodevelopmental deficits. We observed significant reductions in PGC-1α levels across all examined brain structures, as well as a decrease in NRF1 levels in the cerebral cortex and cerebellum of Pb-exposed rats. Additionally, we detected down-regulation of the broad spectrum of fusion-related proteins, including Mfn1 and Opa1 in the hippocampus and cerebellum, and Mfn2 in all analysed brain regions. This was accompanied by increased Fis1 levels in the forebrain cortex and cerebellum, along with a marked depletion of Drp1—a key regulator of mitochondrial fission—in all examined brain structures. These molecular changes were reflected in altered mitochondrial ultrastructure, with mitochondria appearing elongated, swollen, or shrunken in Pb-treated animals.

We also observed brain-structure-specific increases in the expression of ETC complex genes: complex I (*mtNd1*) in the forebrain cortex and complexes I (*mtNd1*), III (*mtCyb*), and IV (*mtCo1*) in the cerebellum of Pb-exposed rats. These changes may indicate the activation of a compensatory or alternative regulatory mechanism aimed at sustaining ATP production despite impaired biogenesis signalling.

Despite up-regulation of classical interferon-stimulated genes (*Mx1*, *Ifi44*, and *Sting1*), along with activation of cyclic GMP-AMP synthase (cGAS)-stimulated and synthesis of 2′3′-cyclic GMP-AMP (2′3′-cGAMP), our data did not show evidence of TBK1 activation. This suggests the existence of a TBK1-independent pathway capable of inducing type I interferon (IFN-I) signalling and the expression of IFN-I–regulated genes during perinatal Pb exposure.

Collectively, these findings indicate that mitochondria are a potential primary target of Pb neurotoxicity and may serve as key mediators of Pb-induced neuroinflammation and neuropathology during critical periods of brain development.

Under conditions of cellular stress—such as oxidative stress, energy depletion, or exposure to heavy metals—mitochondrial dynamics are often disrupted. These disruptions can lead to the accumulation of dysfunctional mitochondria, impaired energy production, increased reactive oxygen species (ROS) generation, and the activation of cell death pathways [14,18,55,56]. The present study provides strong evidence that prenatal and neonatal exposure to Pb, even at concentrations previously considered “safe for humans,” is associated with substantial alterations in the expression patterns of proteins involved in mitochondrial fusion, fission, and biogenesis, potentially contributing to neurodegenerative processes.

Here, we demonstrate for the first time that perinatal Pb exposure significantly decreases the levels of key mitochondrial fusion proteins—Mfn1, Mfn2, and Opa1—in a brain-region-specific manner. Although reductions were observed in both Mfn1 and Mfn2 protein levels, the decrease in Mfn2 appeared more pronounced across all analysed brain structures. This may be due to the broader expression pattern of Mfn2 in various tissues and cell types [57]. Additionally, we found decreased levels of the fission protein Drp1 in all brain regions studied. Interestingly, in contrast to the other fusion/fission proteins, Fis1 levels were significantly increased in the cerebral cortex and cerebellum of Pb-exposed rats.

These findings are consistent with previous reports showing that many neurodegenerative conditions are characterized by reduced levels of fusion proteins (Opa1, Mfn1, Mfn2) and Drp1, along with elevated Fis1 expression [58,59,60]. For instance, in neurons from the hippocampus of Alzheimer’s disease patients, reduced expression of Opa1, Mfn1/2, and Drp1, along with increased Fis1 levels, has been observed [61]. The simultaneous downregulation of fusion proteins and Drp1, along with elevated Fis1, in our study suggests that Pb may disrupt mitochondrial dynamics by impairing both fusion and fission mechanisms.

Supporting our findings, studies by Han et al. [62] and Zhang et al. [57] also documented Pb-induced suppression of mitochondrial fusion machinery proteins alongside increased fission markers, resulting in mitochondrial fragmentation. In chicken spleen cells, Pb exposure significantly downregulated levels of Mfn1, Mfn2, and Opa1, while upregulating fission-related Drp1 and Mff [62]. Similarly, in cultured rat hippocampal neurons and PC12 cells, Pb exposure induced ER stress and ubiquitin-mediated degradation of Mfn2 (and decreases in Mfn1), contributing to mitochondrial fragmentation and neuronal function impairment [57]. Mechanistically, the loss of fusion proteins led to declines in mitochondrial bioenergetics, ROS accumulation, and heightened susceptibility to apoptosis. Rescue experiments targeting Mfn2 or alleviating ER stress reversed these effects, highlighting the essential role of fusion proteins in maintaining mitochondrial integrity under toxic stress [57]. Additional studies have emphasized the critical role of Mfn2 in preserving mitochondrial homeostasis, with its dysfunction linked to the breakdown of mitochondrial networks and multiple neurodegenerative conditions [63].

Yang et al. also reported that ROS-dependent activation of Drp1 in SH-SY5Y neuroblastoma cells led to excessive mitochondrial fragmentation, mitochondrial membrane potential loss, and bioenergetics dysfunction [64]. Enhanced mitochondrial fragmentation has been similarly observed in Pb-exposed SH-SY5Y cells, contributing to increased cell death [65]. Excessive mitochondrial fission disrupts oxidative phosphorylation, elevates oxidative stress, and may ultimately trigger cell death [66,67,68].

In various neuronal injury models, including ischemic stroke and nitric oxide-induced oxidative stress, mitochondrial fission occurs early and contributes to neuronal cell death. Inhibition of fission in these contexts has been shown to offer neuroprotection [69,70]. Blocking Drp1 activity reduces fission and associated neurotoxicity [66], while also preventing Bax translocation to the mitochondria and cytochrome c release—key events in apoptosis [71]. These findings suggest that mitochondrial fragmentation may be a key contributor to Pb-induced neurotoxicity.

Interestingly, in contrast to the majority of data, a study by Dąbrowska et al. showed that treatment of N27 dopaminergic cells with 100 μM Pb for 48 h induced a shift toward mitochondrial fusion. *Mfn2* mRNA levels were increased, while *Drp1* expression was significantly decreased. *Fis1* transcript levels remained unchanged [59]. These discrepancies may reflect differences in exposure models, concentrations, cell types, or the time point of analysis.

As mentioned above, the translocation of Drp1, the key executor of mitochondrial fission, is mediated through its interaction with outer mitochondrial membrane (OMM) adaptor proteins such as Fis1. Together, these proteins form the core of the mitochondrial fission machinery. Therefore, the decreased Drp1 levels observed in our study suggest a reduced capacity of mitochondria to undergo proper division, even in the presence of elevated Fis1. It is worth considering whether the increase in Fis1 concentration may represent a compensatory cellular response to reduced Drp1. However, in the absence of sufficient functional Drp1, mitochondrial fission is likely to remain incomplete or defective.

Although the downregulation of Opa1, Mfn1, and Mfn2, or the overexpression of Fis1, typically promotes mitochondrial fragmentation, the reduction in Drp1 may instead favour elongation. Thus, the net effect on mitochondrial morphology depends on the relative balance between these opposing forces [72]. When both fusion and fission are impaired, mitochondrial morphology may become fragmented if fusion deficits and Fis1 overexpression dominate, elongated if Drp1 reduction is predominant, or exhibit a heterogeneous appearance—with both elongated and fragmented mitochondria—if these processes are similarly affected. In our Pb-exposure model, synaptic mitochondria were observed to be elongated, swollen, or shrunken, suggesting that Drp1 reduction plays a dominant role [73].

Insufficient Drp1 may lead to abnormal elongation and irregular mitochondrial shapes due to defective fission. Impaired fission also hampers mitophagy, preventing the elimination of damaged mitochondrial fragments and leading to their accumulation. In the study by Fowler et al. [74], knockdown of the ER-shaping protein Arl6IP1 reduced Drp1 protein levels, resulting in fewer ER-mitochondrial contacts and impaired mitochondrial loading at the distal ends of long motor neurons. Enhancing fission via overexpression of wild-type Drp1 increased ER-mitochondrial contacts, restored mitochondrial transport within axons, and partially rescued locomotor deficits in a hereditary spastic paraplegia (HSP) model (SPG61) [74]. In adult neurons, the tight regulation of Drp1-dependent fission is essential for mitochondrial bioenergetics and synaptic function [75].

The essential role of the mitochondrial fusion/fission machinery in synapse and dendritic spine formation is well established. Inhibition of mitochondrial fragmentation leads to reduced mitochondrial localization within dendritic spines and impairs synapse development. Furthermore, the absence of Drp1 disrupts the delivery of mitochondria to synapses [76,77].

Mitochondrial dynamics are also strongly dependent on the cytoskeletal network, including microtubules and the actin cytoskeleton. Disruption of mitochondrial transport—regulated by the neuronal cytoskeleton [78]—or imbalance in fission and fusion cycles [77], can impair neurotransmitter release, vesicle recycling, and synaptic plasticity. In our previous study, we observed Pb-induced alterations in Tau, a microtubule-associated protein, indicating neuronal cytoskeleton dysfunction [79], which may explain the abnormalities in mitochondrial fusion/fission processes.

Several biochemical studies have shown that neurons expressing hyperphosphorylated Tau or other pathological protein variants exhibit disrupted fission/fusion balance, resulting in structural and functional mitochondrial abnormalities and ultimately, neuronal damage [80,81,82,83,84]. Therefore, the accumulation of hyperphosphorylated Tau at Ser396 and Ser199/202 in Pb-exposed offspring, previously reported in our studies [75], may partly explain the impaired balance between mitochondrial fission and fusion, contributing to mitochondrial dysfunction and the synaptic pathology observed in our Pb-exposure model. Indeed, we previously identified pathological changes in the ultrastructure of nerve terminals together with altered levels of key synaptic proteins, as a result of perinatal exposure to low doses of Pb [73].

Another emerging yet relatively underexplored issue is the impact of Pb exposure on mitochondrial biogenesis. Research directly examining the effects of Pb on key mitochondrial biogenesis regulators—PGC-1α, TFAM, and NRF1—remains limited. Mitochondrial biogenesis is a tightly regulated process essential for cellular energy homeostasis, involving coordinated transcriptional programs from both nuclear and mitochondrial genomes. A master regulator of this process is peroxisome proliferator-activated receptor gamma coactivator 1-alpha (PGC-1α), which activates nuclear respiratory factors NRF1 and NRF2. These, in turn, stimulate the expression of mitochondrial transcription factor A (TFAM), a key protein responsible for mitochondrial DNA transcription and replication [85].

In the present study, we demonstrated that perinatal Pb exposure adversely affects mitochondrial biogenesis, leading to a significant reduction in PGC-1α levels across all analysed brain regions, as well as NRF1 depletion in the cerebral cortex and cerebellum. These findings align with the growing body of evidence emphasizing the pivotal role of PGC-1α as a master regulator of both mitochondrial biogenesis and dynamics [59]. Dąbrowska et al. [59] also highlighted the neuroprotective role of PGC-1α in counteracting Pb-induced neurotoxicity, particularly by regulating mitochondrial dynamics. Interestingly, contrary to our findings, the study, conducted in substantia nigra cells exposed to 100 μM Pb for 48 h, showed an adaptive response characterized by the upregulation of PGC-1α and its downstream targets, suggesting a compensatory mechanism aimed at preserving mitochondrial biogenesis during acute exposure. However, both insufficient and excessive expression of PGC-1α may disrupt mitochondrial homeostasis under Pb toxicity [59].

Beyond its role in energy metabolism, PGC-1α is also critical for mitochondrial function and regulation of reactive oxygen species (ROS), playing a key role in maintaining redox balance under inflammatory conditions. Reduced PGC-1α activity leads to downregulation of antioxidant gene expression and promotes oxidative stress. This, in turn, may activate pro-inflammatory pathways, including NF-κB, thereby amplifying neuroinflammatory responses [85].

Interaction between PGC-1α and specific transcription factors orchestrates major mitochondrial functions, including fatty acid β-oxidation (FAO), the tricarboxylic acid cycle (TCA), mitochondrial DNA replication, oxidative phosphorylation, and the electron transport chain (OXPHOS/ETC), in addition to organelle biogenesis [86]. In the context of energy metabolism, the PGC-1α–NRF-1/2 axis promotes the transcription of genes encoding mitochondrial complexes I–IV and cytochrome c, thereby facilitating OXPHOS [87]. Consequently, a significant decrease in the expression levels of PGC-1α and NRF1 following Pb exposure would be expected to impair the transcriptional activation of genes critical for oxidative metabolism and mitochondrial maintenance.

When PGC-1α and NRF1 levels decline, the expression of many NRF1-dependent genes, including those encoding essential ETC enzymes, typically decreases in parallel [88]. Paradoxically, however, in the present study, we observed an increase in the expression of genes encoding complexes I, III, and IV of the ETC. This discrepancy suggests that a compensatory or alternative regulatory mechanism may be engaged, potentially aimed at preserving ATP production despite impaired biogenesis signalling. Such compensatory upregulation likely reflects a cellular response to metabolic stress and damage, with the goal of maintaining oxidative phosphorylation capacity.

Nonetheless, this imbalance between transcriptional control and respiratory complex expression may lead to inefficient energy production, excessive reactive oxygen species (ROS) generation, and heightened susceptibility to mitochondrial dysfunction. Our previous studies on cerebellar neurons isolated from the offspring of rats exposed prenatally to subclinical Pb doses confirmed bioenergetics impairment, including significant reductions in intracellular ATP levels, elevated intracellular and mitochondrial ROS, and decreased inner mitochondrial membrane potential [13]. These findings strongly support the notion that the increased expression of ETC complexes I, III, and IV may not translate into functional respiratory chain activity.

Such ETC dysfunction is typically characterized by impaired electron transfer and defective proton gradient formation, directly limiting ATP synthesis and disrupting cellular energy homeostasis [89,90,91]. Enhanced ROS production is a hallmark of ETC abnormalities, as electrons leak from malfunctioning complexes—especially I and III—leading to oxidative damage and further mitochondrial compromise [92,93]. A reduced membrane potential reflects impaired proton motive force, which is essential for ATP synthase activity and overall mitochondrial integrity [91]. These pathological features contribute to cellular dysfunction and are frequently associated with neurodegenerative diseases and other conditions linked to mitochondrial impairment [94,95].

Thus, even in the presence of elevated ETC gene expression, if the corresponding complexes are dysfunctional or improperly assembled, cells may still fail to generate sufficient ATP or maintain redox balance, culminating in the bioenergetics and oxidative stress disturbances observed in our model [89,91].

Maintaining homeostasis in eukaryotic cells critically depends on the continuous availability of ATP, which is generated through glycolysis and oxidative phosphorylation. A comparison of the energy yield from these two pathways underscores the dominant role of mitochondrial oxidative metabolism in meeting the high energy demands of multicellular organisms. Continuous ATP production in mitochondria is primarily supported by the oxidation of metabolites within the tricarboxylic acid (TCA) cycle and fatty acid β-oxidation (FAO), both occurring in the mitochondrial matrix. These catabolic pathways regenerate the reduced nucleotide cofactors NADH and FADH_2_, which serve as electron donors to complex I and complex II of the electron transport chain (ETC), respectively. Electron transfer along the ETC components drives the reduction in oxygen and is coupled with proton translocation across the inner mitochondrial membrane, generating an electrochemical proton gradient between the matrix and the intermembrane space [96]. A by-product of this process is the formation of reactive oxygen species (ROS) [97]. The return flow of protons through complex V (ATP synthase) subsequently drives ADP phosphorylation, thereby ensuring a sustained cellular energy supply.

Numerous studies have demonstrated that exposure to Pb significantly disrupts cellular bioenergetics, most notably through a diminished capacity for ATP production. The mechanisms underlying these energy deficits include inhibition of ETC function, disruption of key enzymes involved in glucose metabolism, inhibition of Na^+^/K^+^-ATPase activity, and a reduction in mitochondrial membrane potential [98,99,100]. Our previous studies on cerebellar neurons isolated from the offspring of rats prenatally exposed to subclinical Pb doses confirmed such mitochondrial dysfunction, including a substantial decline in intracellular ATP levels, inhibition of Na^+^/K^+^-ATPase, increased intracellular and mitochondrial ROS accumulation, and a marked reduction in the inner mitochondrial membrane potential [13].

As outlined, Pb-induced mitochondrial impairment may contribute to the initiation and propagation of neuroinflammatory processes. Neuroinflammation is one of the proposed mechanisms underlying Pb neurotoxicity [50,101,102,103,104,105]. Our previous findings revealed a proinflammatory effect of perinatal Pb exposure, marked by significant activation of both microglia and astrocytes. This gliosis was accompanied by upregulation of classical proinflammatory cytokines, including interleukin-1β (IL-1β), interleukin-6 (IL-6), and tumor necrosis factor-alpha (TNF-α). Importantly, these inflammatory mediators may drive the conversion of astrocytes to the neurotoxic A1 phenotype, exacerbating neuronal injury [50].

A growing body of evidence supports a strong, bidirectional relationship between mitochondrial dysfunction and neuroinflammation. Impaired mitochondrial bioenergetics and elevated oxidative stress act as upstream triggers of inflammatory signalling, while sustained inflammation further disrupts mitochondrial function, promotes organelle damage, and exacerbates oxidative stress, forming a deleterious feed-forward loop [106,107,108,109]. This vicious cycle of neurotoxic events leads to the release of mitochondrial damage-associated molecular patterns (mtDAMPs), which activate specific proinflammatory signalling cascades [110].

In this study, we aimed to analyze the involvement of cGAS–STING signalling in Pb-induced neuroinflammation and neuropathology. Elevated circulating levels of mtDAMPs, particularly mitochondrial DNA (mtDNA), can stimulate the cytosolic DNA-sensing cGAS–STING pathway [48,111]. The release of mtDNA into the cytosol by damaged or dysfunctional mitochondria serves as a key trigger for cGAS-dependent immune activation. The production of the second messenger 2′,3′-cGAMP by cGAS thus links mitochondrial dysfunction to inflammation and neurodegeneration [31,111].

Once synthesized, 2′,3′-cGAMP binds to the adaptor protein STING (stimulator of interferon genes), which is located on the endoplasmic reticulum (ER) membrane [112]. Upon binding 2′,3′-cGAMP, activated STING translocates to the Golgi apparatus and activates TANK-binding kinase 1 (TBK1). This activation involves TBK1 dimerization and autophosphorylation at Ser172 within its activation loop. TBK1 subsequently phosphorylates interferon regulatory factor 3 (IRF3) [87,113]. Phosphorylated IRF3, a master regulator of type I interferons (IFN-I), translocates to the nucleus and turns on transcription of IFN-I genes [114,115].

Additionally, STING within the Golgi apparatus serves as a scaffold to recruit and activate IκB kinase (IKK). Activated IKK phosphorylates IκBα, an inhibitory protein bound to NF-κB. Phosphorylated IκBα undergoes ubiquitination and proteasomal degradation, thereby liberating NF-κB heterodimers (typically composed of p65 and p50 subunits). These activated NF-κB complexes then translocate to the nucleus, where they initiate transcription of pro-inflammatory cytokines and chemokines, amplifying the inflammatory response [44,116,117,118].

Together, the activation of downstream cGAS–STING effectors, IRF3 and NF-κB, induces the robust transcription of type I interferons and various pro-inflammatory mediators, including IL-6 and TNF-α, as part of the classical NF-κB signalling pathway [119]. Secreted IFN-I then acts on target cells (such as neurons, microglia, and astrocytes) via the interferon-alpha/beta receptor (IFNAR), composed of IFNAR1 and IFNAR2 subunits associated with TYK2 and JAK1, respectively. Activation of TYK2 and JAK1 leads to phosphorylation of STAT1 and STAT2, which, along with IRF9, form the ISGF3 complex [120]. This complex translocates to the nucleus and binds to interferon-stimulated response elements (ISREs), initiating the transcription of hundreds of interferon-stimulated genes (ISGs), including *Mx1*, *Ifi44*, and *Sting1* [121,122,123].

Despite the observed upregulation of classical interferon-stimulated genes (*Mx1*, *Ifi44*, and *Sting1*), our data did not show any significant activation of TBK1, as evidenced by the absence of phosphorylation at Ser172. This discrepancy suggests that the transcriptional induction of these ISGs may occur independently of the canonical cGAS–STING–TBK1–IRF3 axis of IFN-I production. Such a finding is atypical, as TBK1 is a central mediator of classical pathways that lead to IFNα/β synthesis—whether triggered by cGAS–STING, RIG-I, or TLR3/TLR4–TRIF pathways. Based on current understanding, TBK1 is essential for ISGs induction; therefore, in the absence of TBK1 activation, robust ISG expression in response to 2′,3′-cGAMP remains difficult to explain.

However, the literature offers some possible alternative mechanisms that may explain this Pb-induced effect. Our findings can be interpreted in the context of functional redundancy and cooperation between the IKK-related kinases TBK1 and IKKε. The IKKε kinase (IKK epsilon—not to be confused with classical IKKα/β) can, like TBK1, phosphorylate IRF3 and induce IFN-I production [124,125,126,127]. Both kinases are well-established mediators of antiviral immunity and are known to activate IRF3 and IRF7, promoting IFN-I transcription and ISG expression [124,125,126,128]. While TBK1 is typically considered the dominant kinase downstream of cGAS–STING, several studies have shown that IKKε can partially compensate for the absence of TBK1 in certain cellular contexts. Although the full interferon response is more robust when both kinases are active, IKKε alone can mediate partial activation [124,126,129].

In support of this, double knockout models lacking both TBK1 and IKKε display a profound loss of IRF3 activation and IFN-β production, underlining their cooperative role in generating an effective antiviral response [125,129,130]. In the study by Perry et al. [129], macrophages from mice with targeted deletion of the *Tbk1* gene showed abolished or delayed IFN-I responses upon LPS and poly(I:C) stimulation. TBK1 was required for IRF3 activation and IFN-β production in response to poly(I:C), while the IFN response to Sendai virus was intact in *Tbk1*^−/−^ macrophages but defective in *Tbk1*^−/−^ embryonic fibroblasts. Moreover, reconstitution of *Tbk1*^−/−^ embryonic fibroblasts with wild-type IKKε restored IFN responses, whereas a kinase-dead IKKε mutant did not, indicating that IKKε can functionally compensate for TBK1 in certain contexts [129].

Thus, our observations may reflect a scenario in which perinatal Pb exposure induces IKKε-mediated signalling that partially sustains IFN-I induction in the absence of detectable TBK1 activity. This mechanism may account for the upregulation of ISGs such as *Mx1*, *Ifi44*, and *Sting1* despite unaltered TBK1 phosphorylation. Nonetheless, further research is required to delineate the specific contributions of TBK1 and IKKε to innate immune activation and gene regulation under Pb-induced neurotoxic conditions.

Secondly, innate immune sensors such as Toll-like receptors (TLRs), particularly TLR7/8 or TLR9 located in endosomes, may be engaged in response to endogenous unmethylated CpG DNA or RNA from damaged cells, leading to type I interferon production via the TLR7/8–MyD88–IRF7(IRF5) and TLR9–MyD88–IRF7(IRF5) signalling pathways. These routes can partially bypass TBK1, particularly in plasmacytoid dendritic cells (pDCs). TLRs signal through MyD88, which recruits downstream effectors such as IRAK1, IRAK4, and TRAF6, ultimately activating IRF7, and to some extent IRF5—especially in inflammatory monocytes. IRF5 participates in MyD88-dependent pathways initiated by TLR7/8/9 that induce type I IFN genes [127].

Another possible mechanism is epigenetic regulation. Lead exposure exerts strong epigenetic effects, altering gene expression through DNA and histone modifications, as well as changes in microRNA regulation [131,132,133,134,135,136,137]. Pb is well-documented to induce global DNA hypomethylation and disrupt the activity and expression of DNA methyltransferases (DNMTs), leading to broad epigenetic changes [135]. This hypomethylation can dysregulate the expression of many genes, potentially including interferon-stimulated genes (ISGs), through mechanisms independent of TBK1 or classical interferon signalling. Numerous studies support this mechanism, showing that Pb exposure leads to widespread DNA methylation loss across various tissues, including neuronal tissue, both during development and in adulthood [138]. Pb-induced hypomethylation involves suppression of DNMT activity and expression, further contributing to gene expression dysregulation [138,139]. Hypomethylation is associated with increased transcriptional activity, especially at enhancer regions and gene promoters, which may lead to upregulation of genes not typically expressed under baseline conditions [138,140]. Thus, global DNA hypomethylation, altered DNMT expression, modified microRNA profiles, and histone hyperacetylation may loosen chromatin structure, unblocking promoters and permitting transcriptional activation [141,142,143,144]. This epigenetic landscape may help explain the observed upregulation of some ISGs and genes related to mitochondrial biogenesis and dynamics in our model.

Finally, Pb exposure may cause dysfunction of the Golgi apparatus, which could indirectly disrupt STING–TBK1 activation. Proper activation of STING requires its translocation from the endoplasmic reticulum (ER) to the Golgi, where essential post-translational modifications—such as palmitoylation and polymerization—occur to initiate downstream signalling and immune activation. Golgi fragmentation or disassembly impairs STING activation, causing it to remain trapped in the ER and unable to interact with TBK1. Consequently, Pb-induced Golgi dysfunction could indirectly inhibit this signalling pathway and contribute to the impaired TBK1 activation observed in our model.

## 4. Materials and Methods

### 4.1. Animals—In Vivo Model

All animal procedures in this study were conducted in compliance with international standards for the care and use of laboratory animals. Efforts were made to minimise the number of animals used and to reduce potential suffering. The experimental protocol was approved by the Local Ethical Committee for Animal Research at the Pomeranian Medical University in Szczecin (Approval No. 5/2014 of 23.04.2014, annexe 2021) in accordance with Directive 2010/63/EU for the protection of animals used for scientific purposes. The experiments were carried out on Wistar rats (Imp: EPI F) supplied by the Prof. J. Nofer Institute of Occupational Medicine in Łódź, Poland. Laboratory for Research on Medicinal and Veterinary Products in the GMP Quality System (Łódź, Poland), which operates breeding of small rodents with the SPF standard. Animals were maintained in a temperature-controlled facility on a 12-h light/dark cycle in open polycarbonate cages in an enriched environment (plastic shelters, wooden blocks, wood shavings, and cotton pads as nesting material). Every effort was made to minimize the number of animals used and reduce the amount of pain and distress. All treatments were performed between 8:00 and 12:00 in the experimental room. Animals were examined daily for signs of suffering and distress. Six three-month-old female Wistar rats (*n* = 6, body weight 250 ± 20 g) were housed with sexually mature males (2:1) for one week, with ad libitum access to food and water. Animals were maintained in a temperature-controlled facility on a 12-h light/dark cycle. After one week, the males were removed; pregnant females were identified and transferred to the experimental breeding facility for acclimatization, and each pregnant female was housed individually. The females were randomly assigned to either the control or experimental group. Rats in the experimental group (*n* = 3) received 0.1% lead acetate (PbAc) in their drinking water ad libitum, beginning on the first day of pregnancy. The PbAc solution was freshly prepared each day in disposable plastic hydropac bags (Anilab, Poland) using the solid reagent dissolved directly in water at the desired concentration without acidification. Control group females (*n* = 3) received distilled water until offspring weaning. There were no significant differences in fluid intake between the control and experimental groups. The offspring (both male and female) remained with their mothers during lactation. Experimental dams continued to receive PbAc in drinking water throughout the lactation period. Offspring were weaned on postnatal day 21 (PND21) and housed separately in groups of 4 in open polycarbonate cages in an enriched environment. From PND21 to PND28, both control and experimental groups received only distilled water ad libitum. The selected Pb exposure method (0.1% PbAc in drinking water) mimics environmental exposure and is a widely used model for lead toxicity in rodents [145,146]. Our previous research [73,79] demonstrated that this protocol results in blood lead levels (Pb-B) in offspring below the previously defined threshold of 10 µg/dL [147,148]. In the present study, the same exposure was used, and Pb administration was discontinued after weaning to maintain Pb-B levels below this threshold. On PND28, the animals were anaesthetized by isoflurane inhalation and euthanized by decapitation. Blood samples and brain tissues were rapidly collected. The hippocampus, forebrain cortex, and cerebellum were dissected and immediately frozen in liquid nitrogen at −80 °C for subsequent analyses. Whole blood was frozen and stored at −80 °C for Pb analysis. Immediately after formation of clot, blood samples were centrifuged at 1000× *g* for 5 min to separate the serum. Samples from a total of 24 offspring (12 from each group) were randomly selected. No a priori exclusion criteria were defined. Operators were not blinded to the experimental group. No significant differences were observed between female and male offspring in any of the measured parameters (*p* = 0.5, Fisher’s exact test), and therefore, both sexes were included in the analyses. The distribution of males and females between the groups, as well as their body weights (males: 70–106 g; females: 52–85.5 g), did not differ significantly (*p* = 0.5, Fisher’s exact test).

### 4.2. Atomic Absorption Spectroscopy for Pb Determination

Lead content was determined using graphite furnace atomic absorption spectrometry (GFAAS) with a Perkin Elmer 4100 ZL spectrometer equipped with Zeeman background correction (Perkin Elmer, Warsaw, Poland). Brain samples were mineralised at 120 °C for 16 h in a closed Teflon container with 1 mL of 65% HNO_3_. After cooling, 1 mL of 30% H_2_O_2_ was added, and the samples were further mineralised for 24 h under the same conditions. The resulting solutions were diluted with deionised water to a final volume of 10 mL and analysed alongside blank and control samples. Whole blood samples were deproteinised using 65% HNO_3_ and analysed using the same protocol. The detection limit for lead was 0.3 μg/dL.

### 4.3. Transmission Electron Microscopy (TEM) Analysis of Brain Samples

Animals were anaesthetised with Nembutal (80 mg/kg body weight) and perfused via the ascending aorta with 2% paraformaldehyde and 2.5% glutaraldehyde in 0.1 M cacodylate buffer, pH 7.4, at 20 °C (Sigma-Aldrich, Poznań, Poland). Tissue samples for ultrastructural analysis were collected from the hippocampus, forebrain cortex, and cerebellum. The samples were fixed in the same solution for 20 h (in the dark, temperature of both the samples and the fixative solution and 4 °C) and then post-fixed in a mixture of 1% OsO_4_ and 0.8% K_4_[Fe(CN)_6_]. Following dehydration in a graded ethanol series and propylene oxide, the tissues were embedded in Spurr resin. Ultrathin sections (50 nm) were examined using a JEM 1200EX electron microscope (Jeol, Tokyo, Japan).

### 4.4. Quantitative Real-Time Polymerase Chain Reaction (qRT-PCR)

Quantitative analysis of mRNA expression was performed for genes encoding mitochondrial fusion proteins (*Mfn1*, *Mfn2*, *Opa1*), mitochondrial fission proteins (*Drp1*, *Fis1*), mitochondrial biogenesis markers (*Pqargc1*, *Tfam1*, *Nrf1*), electron transport chain (ETC) complex components (*mtNd1*, *mtSdha*, *mtCyb1*, *mtCo1*), and interferon-stimulated genes (*Mx1*, *Ifi44*, *Sting1*) using two-step reverse transcription PCR. Total RNA was extracted using the RNeasy Lipid Tissue Mini Kit (Qiagen, Hilden, Germany). Genomic DNA contamination was removed by DNase I treatment according to the manufacturer’s instructions (Sigma-Aldrich, St. Louis, MO, USA). RNA quantity and purity were assessed spectrophotometrically using a NanoDrop ND-1000 spectrophotometer (NanoDrop Technologies, Wilmington, DE, USA). First-strand cDNA synthesis was carried out using a commercial reverse transcription kit and oligo(dT) primers (Fermentas, Waltham, MA, USA) (Appendix A). qPCR was conducted on an ABI 7500 Fast instrument using Power SYBR Green PCR Master Mix (Applied Biosystems, Waltham, MA, USA). The thermal cycling conditions were as follows: 95 °C for 15 s, followed by 40 cycles of 95 °C for 15 s and 60 °C for 1 min. Melting curve analysis confirmed the amplification of a single PCR product for each target gene. Glyceraldehyde-3-phosphate dehydrogenase (GAPDH) served as the endogenous control. Relative gene expression levels were calculated using the 2^ΔCt^ method and expressed as absolute expression values. All data were subjected to statistical analysis.

### 4.5. Enzyme-Linked Immunosorbent Assay (ELISA)

The concentrations of mitochondrial fusion proteins (Mfn1, Mfn2, Opa1), mitochondrial fission proteins (Drp1, Fis1), mitochondrial biogenesis markers (PGC-1α, TFAM, NRF1), and 2′3′-cyclic GMP-AMP (2′3′-cGAMP) were measured using commercial enzyme-linked immunosorbent assay (ELISA) kits according to the manufacturers’ instructions. The following ELISA kits were used: Mfn1 (cat. no. abx259143, Abbexa, Houston, TX, USA), Mfn2 (cat. no. abx259144, Abbexa, Houston, TX, USA), Opa1 (cat. no. LSF74404, LSBio, Newark, CA, USA), PGC-1α (cat. no. ER6455, FineTest, Wuhan, Hubei, China), TFAM (cat. no. abx549437, Abbexa, Houston, TX, USA), NRF1 (cat. no. A5399, Abcam antibodies.com, Cambridge, UK), 2′3′-cGAMP (cat. no. 502510, Cayman chemical, Ann Arbor, MI, USA).

### 4.6. Immunochemical Determination of Protein Levels (Western Blot Analysis)

Western blot analysis was performed under standard conditions to assess protein levels and phosphorylation status. Brain tissue samples were homogenised, mixed with Laemmli buffer, and denatured at 95 °C for 5 min. Proteins were separated by SDS-PAGE and transferred to nitrocellulose membranes using the standard wet-transfer technique. Membranes were washed in TBS-T (Tris-buffered saline with 0.1% Tween 20, pH 7.6) for 5 min and blocked for 1 h at room temperature with 5% (*w*/*v*) nonfat dry milk in TBS-T to prevent non-specific binding. Primary antibodies were applied overnight at 4 °C in 5% (*w*/*v*) nonfat dry milk in TBS-T: anti-TBK1 (1:1000, cat. no 38066, Cell Signaling, Danvers, MA, USA) and anti-p-TBK1(Ser172) (1:1000, cat. no. 54382, Cell Signaling, Danvers, MA, USA). Membranes were then washed three times in TBS-T and incubated for 1 h at room temperature with anti-rabbit secondary antibody (1:4000). After a final wash, proteins were visualised using enhanced chemiluminescence (ECL) reagents (Amersham Biosciences, Bath, UK). Following detection, membranes were stripped and re-probed with anti-GAPDH antibody to confirm equal loading. Densitometric analysis and molecular weight verification were performed using TotalLab4 software (NonLinear Dynamics Ltd., Newcastle upon Tyne, UK).

### 4.7. Statistical Analysis

The group size was calculated with G*Power 3.1. Software (https://www.psychologie.hhu.de/arbeitsgruppen/allgemeine-psychologie-und-arbeitspsychologie/gpower accessed on 22 March 2021), which calculates the minimum required group size based on the size of the Cohen d effect. Standard assumptions adopted: test power = 0.95, significance level = 0.01, groups of equal size. To minimise the potential risk of litter effect, animals were randomly selected from at least three litters per experimental group. The statistical analysis of data was conducted using GraphPad Prism version 8.3.0 (GraphPad Software, San Diego, CA, USA). Results are presented as mean values ± standard error of the mean (SEM). Data distribution was assessed using the Shapiro–Wilk test. The significant outliers were excluded. Comparisons between the control and Pb-treated groups were made using an unpaired Student’s *t*-test or Fisher’s exact test for normally distributed data, and the Mann–Whitney U test for non-normally distributed data. *p*-values < 0.05 were considered statistically significant. In all analyses, “n” refers to the number of independent biological replicates.

## 5. Conclusions

In summary (Figure 9), our study demonstrates that even low-level Pb exposure during critical periods of development profoundly disrupts mitochondrial dynamics, as evidenced by the downregulation of key fusion (Mfn1, Mfn2, Opa1) and fission (Drp1) regulators, along with the mitochondrial biogenesis factors PGC-1α and NRF1, and a concurrent upregulation of Fis1, which together may contribute to neurodevelopmental deficits. Despite these alterations, we observed a paradoxical increase in the expression of genes encoding ETC complexes I, III, and IV, suggesting a possible compensatory response to mitochondrial stress. Intriguingly, Pb exposure also led to elevated levels of 2′,3′-cGAMP and increased transcription of *Mx1*, *Ifi44*, and *Sting*, while TBK1 activation remained unchanged. Collectively, these findings point to a potential TBK1-independent mechanism underlying IFN-I production and ISGs gene induction, and underscore the complexity and redundancy of innate immune signaling, particularly within the central nervous system in response to perinatal exposure to Pb. Pb intoxication has been shown to stimulate IFN-mediated inflammatory responses through the induction of ISGs; however, the precise molecular mechanisms underlying this effect—and the contribution of Pb-induced mitochondrial dysfunction and pathology to this process—remain unclear and warrant further investigation. Taken together, all these findings underscore the need to re-evaluate what constitutes “acceptable” levels of environmental Pb exposure, particularly for vulnerable populations such as pregnant women and young children. Further mechanistic studies are warranted to elucidate the precise signaling pathways linking low-level Pb exposure to disturbances in mitochondrial biogenesis and dynamics, as well as inflammatory responses, to assess the long-term neurological consequences of such alterations.

## Figures and Tables

**Figure 1 ijms-26-11907-f001:**
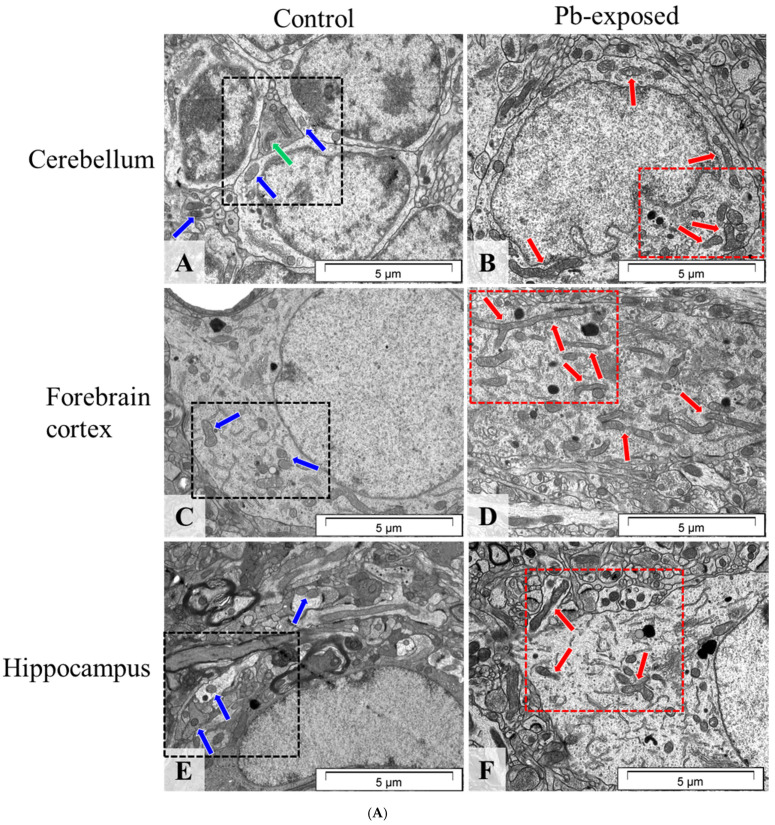
(**A**). Representative transmission electron microscopy images of brain regions in the control (**A**,**C**,**E**) and Pb-exposed (**B**,**D**,**F**) rats. (**A**) Cerebellar granule cells from control rats; blue arrows—normal mitochondria, green arrow—likely dividing mitochondria. (**B**) Cerebellar granule cell from a Pb-exposed rat; red arrows—altered mitochondria, highly elongated, fragmented, or deformed. (**C**) Neuronal cell fragment from the forebrain cortex of control rats; blue arrows—normal mitochondria. (**D**) Nerve cell fragment from the forebrain cortex of Pb-exposed rats; red arrows—elongated, fragmented, or morphologically altered mitochondria. (**E**) Neuronal cell fragment from the hippocampus of control rats; blue arrows—normal mitochondria. (**F**) Neuronal cell fragment from the hippocampus of Pb-exposed rats; red arrows—fragmented and morphologically abnormal mitochondria. Representative pictures from *n* = 4 separate animals for the control and *n* = 4 separate animals for the experimental group were presented. (**B**). Representative transmission electron microscopy images of neural mitochondria in the control (**A’**,**C’**,**E’**) and Pb-exposed (**B’**,**D’**,**F’**) rats in three different areas of the brain: cerebellum (**A’**,**B’**), forebrain cortex (**C’**,**D’**) and hippocampus (**E’**,**F’**). (**A’**) Control cerebellar neuron with normal mitochondria—blue arrows, and likely dividing mitochondria—green arrow. (**B’**) Pb-exposed cerebellar neuron with altered mitochondria, highly elongated, fragmented, or deformed—red arrows. (**C’**) Control forebrain cortical neuron with normal mitochondria—blue arrows. (**D’**) Pb-exposed forebrain cortical neuron with elongated, fragmented, or morphologically altered mitochondria—red arrows. (**E’**) Control hippocampal neuron with normal mitochondria—blue arrows. (**F’**) Pb-exposed hippocampal neuron with fragmented and morphologically abnormal mitochondria—red arrows.

**Figure 2 ijms-26-11907-f002:**
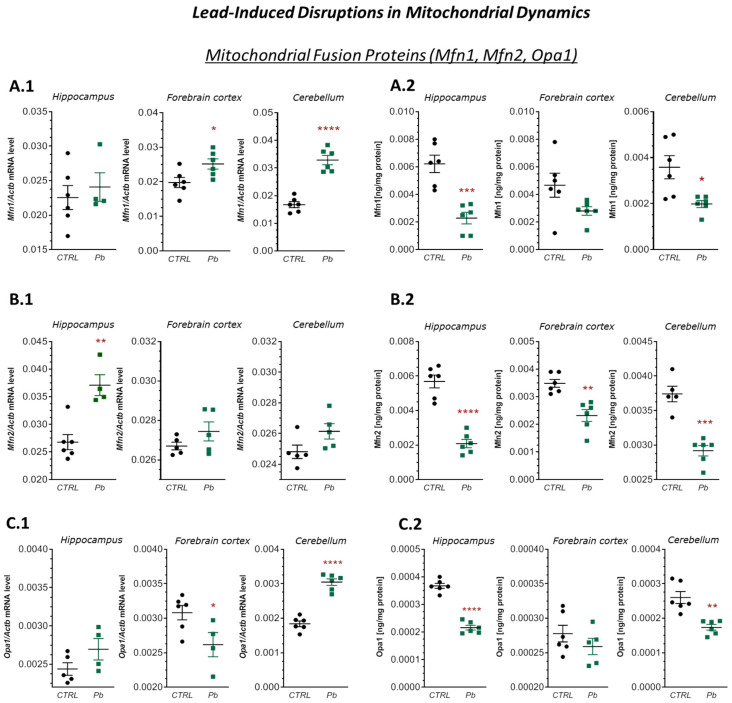
Effects of perinatal Pb exposure on mRNA expression and protein levels of *Mfn1*, *Mfn2*, and *Opa1* in the hippocampus, forebrain cortex, and cerebellum of PND28 rat offspring. (**A.1**–**C.1**)—mRNA levels of *Mfn1*, *Mfn2*, and *Opa1*, respectively, determined by qRT-PCR and normalised to *Actb*. (**A.2**–**C.2**)—Protein concentrations of Mfn1, Mfn2, and Opa1, respectively, measured by ELISA. Data represent the mean ± SEM from *n* = (4–6) independent animals from three different litters. Comparisons between the control (CTRL) and Pb-treated groups (Pb) were made using an unpaired, two-tailed Student’s *t*-test for normally distributed data. * *p* < 0.05, ** *p* < 0.01, *** *p* < 0.001, **** *p* < 0.0001 versus corresponding control.

**Figure 3 ijms-26-11907-f003:**
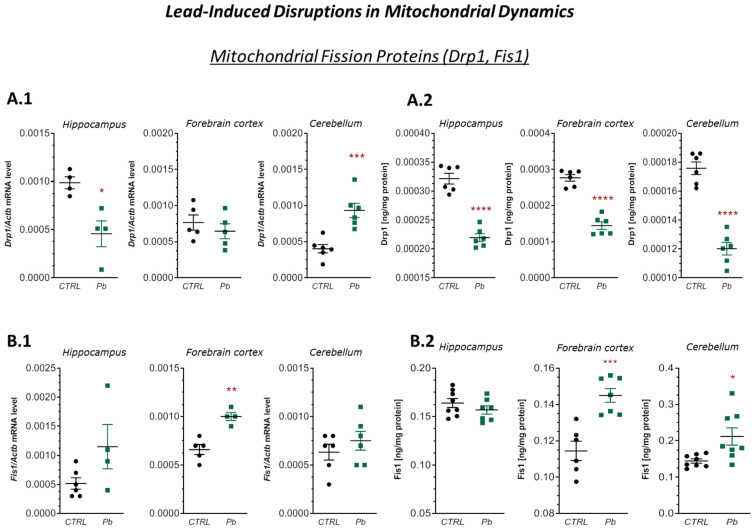
Effects of perinatal Pb exposure on *Drp1* and *Fis1* mRNA expression and protein concentrations in the hippocampus, forebrain cortex, and cerebellum of PND28 rat offspring. (**A.1**,**B.1**)—mRNA levels of *Drp1* and *Fis1* measured by qRT-PCR, normalised to *Actb*. (**A.2**,**B.2**)—Protein concentrations of Drp1 and Fis1 determined by ELISA. Data represent mean ± SEM from *n* = (4–6) independent animals from three different litters. Comparisons between the control (CTRL) and Pb-treated groups (Pb) were made using an unpaired, two-tailed Student’s *t*-test for normally distributed data. * *p* < 0.05, ** *p* < 0.01, *** *p* < 0.001, **** *p* < 0.0001 versus corresponding control.

**Figure 4 ijms-26-11907-f004:**
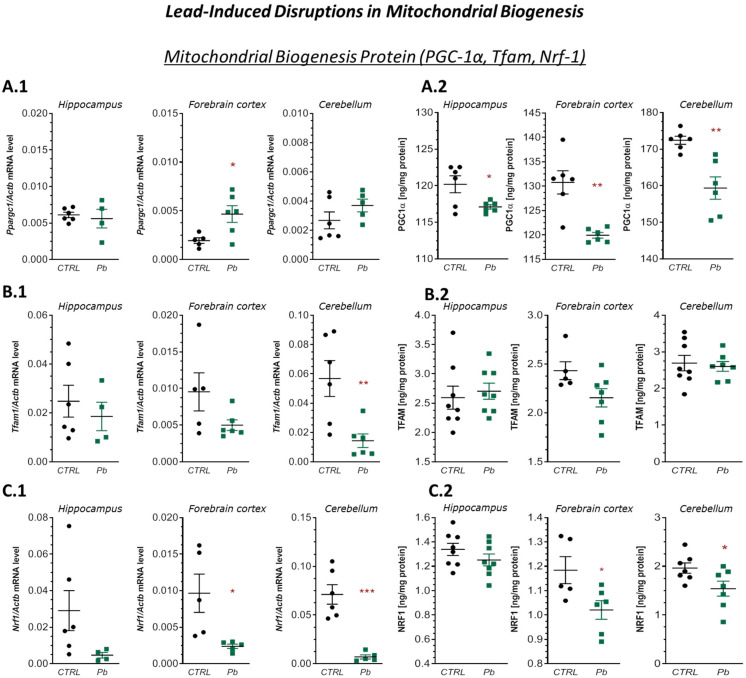
Effects of perinatal Pb exposure on *Pqargc1*, *Tfam1*, and *Nrf1* mRNA expression and corresponding protein levels in the hippocampus, forebrain cortex, and cerebellum of PND28 rat offspring. (**A.1**–**C.1**)—mRNA levels of *Pqargc1*, *Tfam1*, and *Nrf1* measured by qRT-PCR and normalised to *Actb*. (**A.2**–**C.2**)—Protein concentrations of PGC-1α, TFAM, and NRF1 determined by ELISA. Data represent mean ± SEM from *n* = (4–6) independent animals from three different litters. Comparisons between the control (CTRL) and Pb-treated groups (Pb) were made using an unpaired, two-tailed Student’s *t*-test or the Mann–Whitney U test, for normally and non-normally distributed data, respectively. * *p* < 0.05, ** *p* < 0.01, *** *p* < 0.001 versus corresponding control.

**Figure 5 ijms-26-11907-f005:**
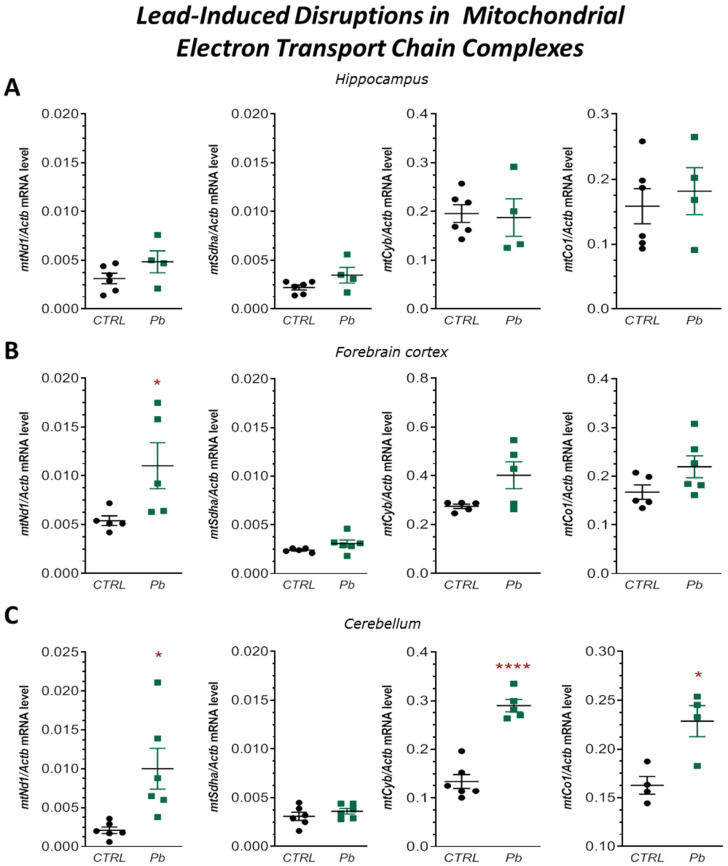
Effects of perinatal Pb exposure on mRNA expression of ETC complex subunits *mtNd1*, *mtSdha*, *mtCyb*, and *mtCo1* in the hippocampus (**A**), forebrain cortex (**B**), and cerebellum (**C**) of PND28 rat offspring. Gene expression was assessed by qRT-PCR and normalized to *Actb*. Data are presented as mean ± SEM from *n* = (4–6) independent animals from three different litters. Comparisons between the control (CTRL) and Pb-treated groups (Pb) were made using an unpaired, two-tailed Student’s *t*-test for normally distributed data. * *p* < 0.05, **** *p* < 0.0001 versus corresponding control.

**Figure 6 ijms-26-11907-f006:**
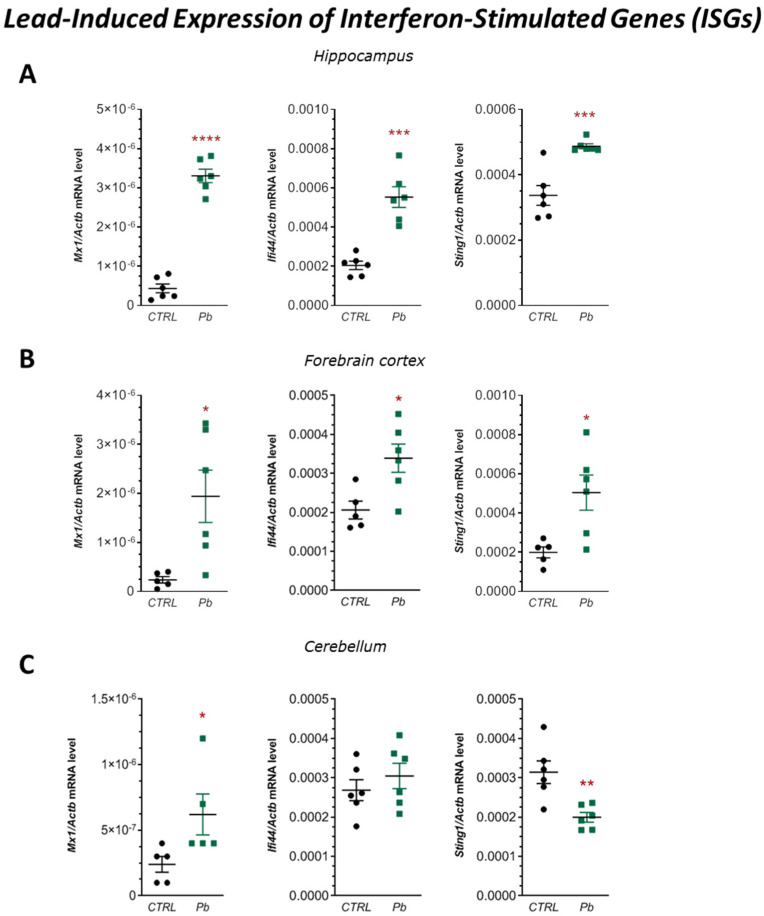
Effects of perinatal Pb exposure on mRNA expression of ISGs: *Mx1*, *Ifi44*, and *Sting1* in the hippocampus (**A**), forebrain cortex (**B**), and cerebellum (**C**) of PND28 rat offspring. Gene expression was measured by qRT-PCR and normalised to *Actb*. Data are presented as mean ± SEM from *n* = (5–6) independent animals from three different litters. Comparisons between the control (CTRL) and Pb-treated groups (Pb) were made using an unpaired, two-tailed Student’s *t*-test or the Mann–Whitney U test, for normally and non-normally distributed data, respectively. * *p* < 0.05, ** *p* < 0.01, *** *p* < 0.001, **** *p* < 0.0001 versus corresponding control.

**Figure 7 ijms-26-11907-f007:**
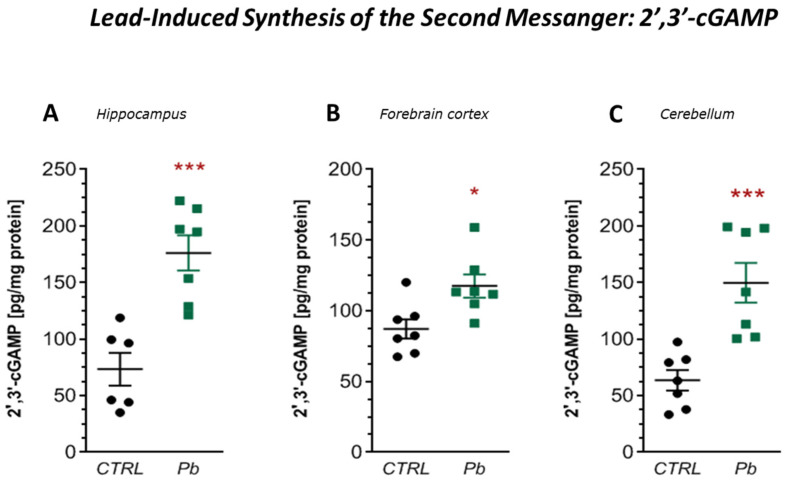
Effects of perinatal Pb exposure on 2′,3′-cGAMP levels in the hippocampus (**A**), forebrain cortex (**B**), and cerebellum (**C**) of PND28 rat offspring. Concentrations were measured by ELISA. Data represent mean ± SEM from *n* = (6–7) independent animals from three different litters. Comparisons between the control (CTRL) and Pb-treated groups (Pb) were made using an unpaired, two-tailed Student’s *t*-test for normally distributed data. * *p* < 0.05, *** *p* < 0.001 versus corresponding control.

**Figure 8 ijms-26-11907-f008:**
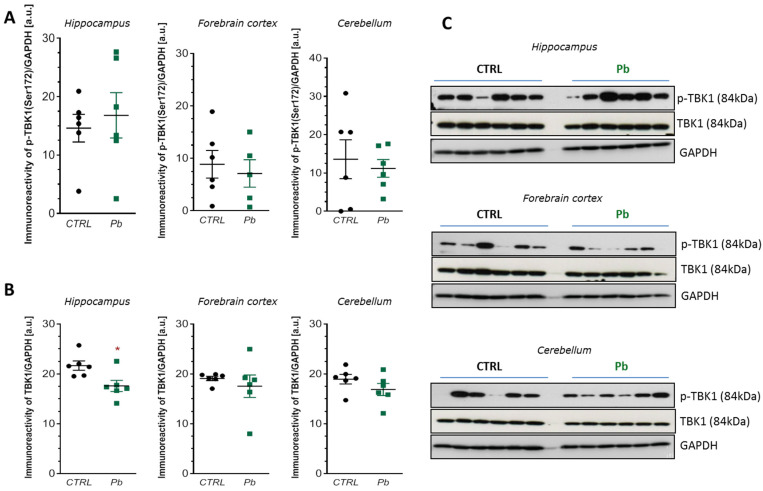
Effects of perinatal Pb exposure on TBK1 activation in the brain of PND28 rat offspring. Immunoreactivity of p-TBK1(Ser172) and TBK1 in control and Pb-affected rats was monitored using Western blot analysis. Densitometric analysis of p-TBK1, phosphorylated at (Ser172) (**A**), and TBK1 (**B**), together with representative pictures (**C**), were shown. Results were normalised to GAPDH. Data represent mean ± SEM from *n* = 6 independent animals from three different litters. Comparisons between the control (CTRL) and Pb-treated groups (Pb) were made using an unpaired, two-tailed Student’s *t*-test for normally distributed data. * *p* < 0.05 versus corresponding control.

**Figure 9 ijms-26-11907-f009:**
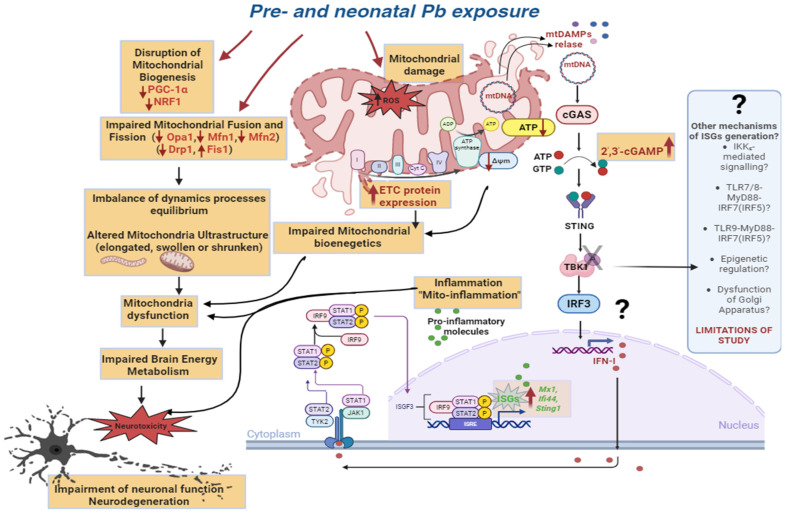
Molecular and cellular mechanisms of the toxic effects of lead (Pb) on mitochondria during the prenatal and neonatal periods. Exposure to Pb induces oxidative stress (ROS) and disrupts mitochondrial biogenesis (↓ PGC-1α, ↓ NRF1). It also impairs mitochondrial fusion and fission (↓ Opa1, ↓ Mfn1, ↓ Mfn2, ↓ Drp1, ↑ Fis1), processes essential for mitochondrial network formation and ultrastructural remodeling (elongation, swelling, or contraction). Altered mitochondrial dynamics lead to dysfunction and impaired energy production (reduced membrane potential (↓ ΔΨ), decreased ATP (↓ ATP) synthesis, and increased expression of ETC complex proteins, likely reflecting a compensatory response). Ultimately, these changes impair cerebral energy metabolism and promote neurotoxicity, resulting in neuronal dysfunction and neurodegeneration. In addition, damaged mitochondria release mtDAMPs, including mtDNA, which activate the cGAS sensor. ATP and GTP are then converted into the secondary messenger 2′,3′-cGAMP, which activates STING. Upon reaching the ERGIC and Golgi compartments, the C-terminal tail of STING recruits downstream TBK1 through a conserved PLPLRT/SD amino acid–binding motif, promoting dimerization-mediated TBK1 autophosphorylation at (Ser172) within the activation loop and thereby initiating TBK1 activation. Activated TBK1 subsequently phosphorylates STING in its C-terminal tail, generating a binding site for IRF3, which is then recruited and phosphorylated by adjacent TBK1. Activation of this pathway drives the expression of type I interferon (IFN-I) and interferon-stimulated genes (ISGs), including *Mx1*, *Ifi44*, and *Sting1*. Once secreted, IFN-I acts through IFNAR on target cells (neurons, microglia, and astrocytes), comprising IFNAR1 and IFNAR2 subunits associated with TYK2 and JAK1, respectively. Activation of TYK2 and JAK1 promotes phosphorylation of STAT1 and STAT2, which heterodimerize with IRF9 to form the ISGF3 complex. ISGF3 translocates to the nucleus, where it binds ISRE-containing genes and stimulates the transcription of numerous ISGs, including pro-inflammatory mediators. This “mito-inflammation” exacerbates mitochondrial dysfunction, further compromising cerebral energy homeostasis. The final outcome is neurotoxicity and neuronal damage. Unresolved mechanisms and possible alternative pathways leading to ISG expression (e.g., *Mx1*, *Ifi44*, *Sting1*) in light of the lack of TBK1 phosphorylation (grey cross) and activation observed in our study include: IKKε-dependent signaling, TLR7/8–MyD88–IRF7/IRF5 and TLR9–MyD88–IRF7/IRF5 pathways, epigenetic regulation, Golgi apparatus dysfunction, which represent limitation of the present study and directions for future research.

## Data Availability

The raw data supporting the conclusions of this article will be made available by the authors on request.

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
