# Peer review of "Disturbances in Mitochondrial Network, Biogenesis, and Mitochondria-Mediated Inflammatory Responses in Selected Brain Structures of Rats Exposed to Lead (Pb) During Prenatal and Neonatal Development"

_ijms, 2025, doi:10.3390/ijms262411907_

Round 1

Reviewer 1 Report

Comments and Suggestions for Authors

1. The abstract should be structured with headings such as "Background," "Methods," "Results," and "Conclusions," and appropriately condensed. Any revisions must strictly adhere to the journal's guidelines. The full term of an abbreviation should be provided upon its first occurrence in the main text.

2. The introduction is overly lengthy and should be streamlined. For instance, experimental results should not be introduced here, and discussions of results should be reserved for the "Discussion" section. Additionally, the logical flow needs improvement, and a reorganization is recommended.

3. In Section 2.1.2, it is advisable to include a possible explanation for the inconsistency between mRNA and protein levels of mitochondrial fusion/fission proteins.

4. In Section 4.3, when stating "samples were fixed in the same solution for 20 hours," the temperature of both the samples and the fixative solution should be specified.

5. The animal license number must be provided. The authors are encouraged to supplement this information.

6. When anesthetizing animals on PND28, the specific anesthetic used should be indicated.

7. Reference formatting should be standardized and primarily include publications from the last five years.

Author Response

Dear Editor,

Dear Reviewers,

We would like to thank you very much for the consideration of our manuscript: “Disturbances in mitochondrial network, biogenesis, and mitochondria-mediated inflammatory responses in selected brain structures of rats exposed to lead (Pb) during prenatal and neonatal development” for publication in the IJMS (Manuscript ID: ijms-3964507). We have taken into consideration very carefully all the remarks and comments of both Reviewers. The manuscript was corrected according to the Reviewer’s remarks (all changes are marked in red colour).

In addition, we wish to inform you that, in accordance with the Assistant Editor's suggestion, we have completed these sections:

  1. Heading Number Format Check.

We have reviewed all headings to ensure correct and sequential numbering.

  1. Suggestions on replacing retraction/correction references.

We have replaced reference 55. Dabrowska A, Luis Venero JL, Iwasawa R, Hankir M, Rahman S, Boobis A, Hajji N. Erratum: PGC-1α controls mitochondrial biogenesis and dynamics in lead-induced neurotoxicity. Aging (Albany NY). 2016 Apr 23; 8:832-832. https://doi.org/10.18632/aging.100955

  1. Completing the Back Matter.

Funding: The Funding section is complete. Due to a lack of a grant, we haven't provided the specific grant number.

  1. Informed Consent Statement.

This statement was added.  The declaration is "Not applicable."

  1. Data Availability Statement: We ensured the inclusion of the full, standard statement: "The raw data supporting the conclusions of this article will be made available by the authors on request."

We consider that all corrections increased the value of our article, and our explanations you will find adequate. I hope our corrected manuscript can be accepted for publication in IJMS.

I would very much appreciate your opinion thereon.

Yours faithfully,

Irena Baranowska-Bosiacka

Answers from the authors of the manuscript to Reviewers’ questions:

Answer to Reviewer 1

Reviewer #1:

  1. The abstract should be structured with headings such as "Background," "Methods," "Results," and "Conclusions," and appropriately condensed. Any revisions must strictly adhere to the journal's guidelines. The full term of an abbreviation should be provided upon its first occurrence in the main text.
  2. The introduction is overly lengthy and should be streamlined. For instance, experimental results should not be introduced here, and discussions of results should be reserved for the "Discussion" section. Additionally, the logical flow needs improvement, and a reorganization is recommended.
  3. In Section 2.1.2, it is advisable to include a possible explanation for the inconsistency between mRNA and protein levels of mitochondrial fusion/fission proteins.
  4. In Section 4.3, when stating "samples were fixed in the same solution for 20 hours," the temperature of both the samples and the fixative solution should be specified.
  5. The animal license number must be provided. The authors are encouraged to supplement this information.
  6. When anesthetizing animals on PND28, the specific anesthetic used should be indicated.
  7. Reference formatting should be standardized and primarily include publications from the last five years.

Answer: We are very grateful for the examination of our manuscript and for the valuable comments, and also for the time offered for the preparation of the revision. We fully agree with your opinion. We have carefully taken into consideration all remarks and comments. The manuscript was improved according to your suggestions. All changes to the manuscript are highlighted. We believe that these changes have resulted in a substantially strengthened manuscript for potential publication in IJMS. Please find below our response.

Point 1 The abstract should be structured with headings such as "Background," "Methods," "Results," and "Conclusions," and appropriately condensed. Any revisions must strictly adhere to the journal's guidelines. The full term of an abbreviation should be provided upon its first occurrence in the main text.

Answer: We would like to thank the Reviewer for this valuable comment. We fully agree with your opinion, so we structured the abstract. However, taking into account the Journal's guidelines: ("The abstract should be a single paragraph and should follow the style of structured abstracts, but without headings: 1) Background 2) Methods 3) Results and 4) Conclusion"), we structured the Abstract without headings such as "Background", "Methods", "Results" and "Conclusions" in the text.

Below, I am attaching a shortened abstract after major revision:

“Pb disrupts mitochondrial function, but its impact on the mitochondrial dynamics and biogenesis during early brain development remains insufficiently understood. This study aimed to investigate the effects of pre- and neonatal Pb exposure on the processes involved in mitochondrial network formation in the brains of rat offspring, simulating environmental exposure. We quantified mRNA expression (qRT-PCR) and protein levels (ELISA) of key mitochondrial fusion (Mfn1, Mfn2, Opa1), fission (Drp1, Fis1) regulators, as well as biogenesis markers (PGC-1α, TFAM, NRF1) in the hippocampus, forebrain cortex, and cerebellum of rats exposed to Pb. Mitochondrial ultrastructure was evaluated using transmission electron microscopy (TEM), and the expression of mitochondrial electron transport chain (ETC) genes was analysed using qRT-PCR. Furthermore, to examine the involvement of the cGAS–STING pathway in Pb-induced neuroinflammation, we measured the expression of ISGs (qRT-PCR), TBK1 phosphorylation (Western blot), and 2′,3′-cGAMP synthesis (ELISA). Our results showed that Pb exposure markedly reduced PGC-1α and region-specific NRF1 levels, broadly suppressed fusion proteins (Mfn1, Mfn2, Opa1), increased Fis1, and depleted Drp1. ETC gene expression (mtNd1, mtCyb and mtCo1) were upregulated in a brain-structure-dependent manner. These molecular changes were accompanied by pronounced mitochondrial morphological abnormalities. Despite upregulation of Mx1, Ifi44, and Sting1, along with synthesis of 2′3′-cGAMP, TBK1 activation was not detected. All these findings demonstrate that early-life Pb exposure, even low-dose, disrupts mitochondrial biogenesis and the fusion–fission machinery, thus impairs brain energy homeostasis, and implicates mitochondria as central mediators of Pb-induced neuroinflammation and neurodevelopmental toxicity”.

Point 2 The introduction is overly lengthy and should be streamlined. For instance, experimental results should not be introduced here, and discussions of results should be reserved for the "Discussion" section. Additionally, the logical flow needs improvement, and a reorganization is recommended.

Answer: We would like to thank the Reviewer for this valuable comment. We fully agree with your opinion, so we shortened and modified the Introduction according to your suggestion. All changes to the Manuscript are highlighted.

Point 3 In Section 2.1.2, it is advisable to include a possible explanation for the inconsistency between mRNA and protein levels of mitochondrial fusion/fission proteins.

Answer: Thank you very much for these remarks. We fully agree with your opinion. According to this, we have included a possible explanation for the inconsistency between mRNA and protein levels.

Below, I am attaching explanation attached to the manuscript:

A possible explanation for the inconsistency between mRNA and protein levels of mitochondrial fusion/fission proteins observed in our studies may result from the effect of Pb on post-transcriptional processes through translation inhibition [50], miRNA-mediated regulation [51], and altered mRNA stability/degradation [52]. These processes are prominent in mainly neuronal models, where Pb-induced ROS and Ca²⁺ dysregulation amplify post-transcriptional control, leading to attenuated protein responses during neurotoxicity [53].

Detailed explanation of the mechanisms:

Post-transcriptional processes cause mRNA-protein mismatches after Pb exposure through translation inhibition, miRNA-mediated regulation, and altered mRNA stability/degradation.​

Key Mechanisms and Evidence from Studies:

1) Translation inhibition: Pb phosphorylates eIF2α via PERK activation from ER stress and binds ribosomes, blocking mRNA recruitment and initiation of protein synthesis despite elevated mRNA levels (e.g., for synaptic proteins like synaptophysin).​
Virgolini, M.B., Aschner, M. Molecular mechanisms of lead neurotoxicity. Adv Neurotoxicol. 2021, 17;5:159–213. doi: 10.1016/bs.ant.2020.11.002.

2) miRNA induction and silencing: Pb upregulates miRNAs (e.g., miR-34a, miR-29 family) that bind 3'UTRs of target mRNAs, promoting deadenylation, decapping, or translational repression, reducing protein output from stress-response transcripts. Mei, Z., Liu, G., Zhao, B., He, Gu, S. Emerging roles of epigenetics in lead-induced neurotoxicity. Environ. Int. 2023,  Nov:181:108253. doi: 10.1016/j.envint.2023.108253.

3) mRNA destabilization and decay: Pb enhances nonsense-mediated decay (NMD) or AU-rich element-mediated decay for specific mRNAs (e.g., DNA repair genes like OGG1), decoupling steady-state mRNA from protein accumulation. Hernández-Elvira, M. Sunnerhagen, P.  Post-transcriptional regulation during stress. FEMS Yeast Res. 2022, 22(1):foac025. doi: 10.1093/femsyr/foac025.

These processes are prominent in mainly neuronal models, where Pb-induced ROS and Ca²⁺ dysregulation amplify post-transcriptional control, leading to attenuated protein responses during neurotoxicity. Morgan, R.K.  Tapaswi, A.,  Polemi, K.M.,  Tolrud,  E.C., Bakulski, K.M, Svoboda, L.K., Dolinoy, D.C, Colacino, J.A. Environmentally Relevant Lead Exposure Impacts Gene Expression in SH-SY5Y Cells Throughout Neuronal Differentiation. Toxicological Sci. 2025, 207(2):435-448. doi: 10.1093/toxsci/kfaf072.

Point 4 In Section 4.3, when stating "samples were fixed in the same solution for 20 hours," the temperature of both the samples and the fixative solution should be specified.

Answer: Thank you very much for these remarks. According to your request, the temperature of both the samples and the fixative solution has been specified. Below, I am attaching explanation attached to the manuscript:

“Tissue samples for ultrastructural analysis were collected from the hippocampus, forebrain cortex, and cerebellum. The samples were fixed in the same solution for 20 hours (in the dark, temperature of both the samples and the fixative solution and 4oC) and then post-fixed in a mixture of 1% OsO₄ and 0.8% K₄[Fe(CN)₆]”.

Point 5 The animal license number must be provided. The authors are encouraged to supplement this information.

Answer: Thank you very much for these remarks. According to your request, the animal license number has been provided. Below, I am attaching explanation attached to the manuscript:

“The experiments were carried out on Wistar rats (Imp: EPI F) supplied by the Prof. J. Nofer Institute of Occupational Medicine in Łódź. Laboratory for Research on Medicinal and Veterinary Products in the GMP Quality System (Łódź, Poland), which operates breeding of small rodents with the SPF standard”.

Point 6 When anesthetizing animals on PND28, the specific anesthetic used should be indicated.

Answer: Thank you very much for these remarks. According to your request, an anaesthetic was indicated. Below, I am attaching explanation attached to the manuscript:

“On PND28, the animals were anaesthetized by isoflurane inhalation and euthanized by decapitation”.

Point 7 Reference formatting should be standardized and primarily include publications from the last five years.

Answer: Thank you very much for these remarks. Reference formatting has been standardised. In our opinion, the publications older than five years can certainly be scientifically valuable, and there is no general rule prohibiting citing them. In many fields, classic, older sources are the foundation of knowledge.

Publications older than five years still hold significant scientific value and should not be excluded from bibliographies solely because of their publication date. In many fields, classic works form the foundation of contemporary knowledge, and their concepts, definitions, and models remain relevant despite the passage of time. Older sources also allow for tracing the development of scientific thought and situating the latest research in a broader theoretical context. Therefore, in scientific work, it is advisable to consult both current publications and older works that have retained their substantive value and significance for a given discipline.

We would therefore like to kindly ask you to acknowledge previous studies that are relevant to explaining our observations.

Reviewer 2 Report

Comments and Suggestions for Authors

The manuscript addresses an important topic regarding the impact of lead exposure on mitochondrial dynamics and innate immune signaling in the brain. There are several points should be addressed/improved before publication.

  1. mRNA–protein discordance not explained well. Multiple key genes show opposite regulation at the mRNA and protein level. So additional experiments (e.g., protein stability, translational regulation) or mechanistic explanation/discussion should be provided.
  2. The authors mainly use ELISA for quantify mitochondrial proteins. Other methods should be provided as alternative ways, such as Western blotting, or quantitative proteinomics, particularly for the gene showing opposite regulation at protein and mRNA level.
  3. For figure 8, the level of pTBK1 alter dramatically in different brains. So "TBK1-independent IFN-I pathway" is not conclusively. more samples should be tests. Were these samples collected or processed differently to cause the big changes?
  4. The resolution of Figure 1 should be increase, or provide zoom-in images to show mitochondria.
  5. Table1 should be provided as a supplemental file.
  6. the y axis of qPCR graph should be presented as scientific natation way instead with decimal notation (such the scale of 0.000001)

Author Response

Dear Editor,

Dear Reviewers,

We would like to thank you very much for the consideration of our manuscript: “Disturbances in mitochondrial network, biogenesis, and mitochondria-mediated inflammatory responses in selected brain structures of rats exposed to lead (Pb) during prenatal and neonatal development” for publication in the IJMS (Manuscript ID: ijms-3964507). We have taken into consideration very carefully all the remarks and comments of both Reviewers. The manuscript was corrected according to the Reviewer’s remarks (all changes are marked in red colour).

In addition, we wish to inform you that, in accordance with the Assistant Editor's suggestion, we have completed these sections:

  1. Heading Number Format Check.

We have reviewed all headings to ensure correct and sequential numbering.

  1. Suggestions on replacing retraction/correction references.

We have replaced reference 55. Dabrowska A, Luis Venero JL, Iwasawa R, Hankir M, Rahman S, Boobis A, Hajji N. Erratum: PGC-1α controls mitochondrial biogenesis and dynamics in lead-induced neurotoxicity. Aging (Albany NY). 2016 Apr 23; 8:832-832. https://doi.org/10.18632/aging.100955

  1. Completing the Back Matter.

Funding: The Funding section is complete. Due to a lack of a grant, we haven't provided the specific grant number.

  1. Informed Consent Statement.

This statement was added.  The declaration is "Not applicable."

  1. Data Availability Statement: We ensured the inclusion of the full, standard statement: "The raw data supporting the conclusions of this article will be made available by the authors on request."

We consider that all corrections increased the value of our article, and our explanations you will find adequate. I hope our corrected manuscript can be accepted for publication in IJMS.

I would very much appreciate your opinion thereon.

Yours faithfully,

Irena Baranowska-Bosiacka

Answer to Reviewer 2

Reviewer #2: The manuscript addresses an important topic regarding the impact of lead exposure on mitochondrial dynamics and innate immune signaling in the brain. There are several points should be addressed/improved before publication.

  1. mRNA–protein discordance not explained well. Multiple key genes show opposite regulation at the mRNA and protein level. So additional experiments (e.g., protein stability, translational regulation) or mechanistic explanation/discussion should be provided.
  2. The authors mainly use ELISA for quantify mitochondrial proteins. Other methods should be provided as alternative ways, such as Western blotting, or quantitative proteinomics, particularly for the gene showing opposite regulation at protein and mRNA level.
  3. For figure 8, the level of pTBK1 alter dramatically in different brains. So "TBK1-independent IFN-I pathway" is not conclusively. more samples should be tests. Were these samples collected or processed differently to cause the big changes?
  4. The resolution of Figure 1 should be increase, or provide zoom-in images to show mitochondria.
  5. Table1 should be provided as a supplemental file.
  6. the y axis of qPCR graph should be presented as scientific natation way instead with decimal notation (such the scale of 0.000001)

Answer: Thank you very much for the examination of our Manuscript and for the valuable comments, and also for the time offered for the preparation of the revision. We have carefully taken into consideration all comments.

Point 1 mRNA–protein discordance not explained well. Multiple key genes show opposite regulation at the mRNA and protein level. So additional experiments (e.g., protein stability, translational regulation) or mechanistic explanation/discussion should be provided.

Answer: We would like to thank the Reviewer for this valuable comment. We fully agree with your opinion, so we provided a possible explanation for the inconsistency between mRNA and protein levels in the section Results 2.2.2. Perinatal Pb exposure affected mitochondrial dynamics in the brain of rat offspring.

Below, I am attaching explanation attached to the manuscript:

A possible explanation for the inconsistency between mRNA and protein levels of mitochondrial fusion/fission proteins observed in our studies may result from the effect of Pb on post-transcriptional processes through translation inhibition [50], miRNA-mediated regulation [51], and altered mRNA stability/degradation [52]. These processes are prominent in mainly neuronal models, where Pb-induced ROS and Ca²⁺ dysregulation amplify post-transcriptional control, leading to attenuated protein responses during neurotoxicity [53].

Detailed explanation of the mechanisms:

Post-transcriptional processes cause mRNA-protein mismatches after Pb exposure through translation inhibition, miRNA-mediated regulation, and altered mRNA stability/degradation.​

Key Mechanisms and Evidence from Studies:

1) Translation inhibition: Pb phosphorylates eIF2α via PERK activation from ER stress and binds ribosomes, blocking mRNA recruitment and initiation of protein synthesis despite elevated mRNA levels (e.g., for synaptic proteins like synaptophysin).​
Virgolini, M.B., Aschner, M. Molecular mechanisms of lead neurotoxicity. Adv Neurotoxicol. 2021, 17;5:159–213. doi: 10.1016/bs.ant.2020.11.002.

2) miRNA induction and silencing: Pb upregulates miRNAs (e.g., miR-34a, miR-29 family) that bind 3'UTRs of target mRNAs, promoting deadenylation, decapping, or translational repression, reducing protein output from stress-response transcripts. Mei, Z., Liu, G., Zhao, B., He, Gu, S. Emerging roles of epigenetics in lead-induced neurotoxicity. Environ. Int. 2023,  Nov:181:108253. doi: 10.1016/j.envint.2023.108253.

3) mRNA destabilization and decay: Pb enhances nonsense-mediated decay (NMD) or AU-rich element-mediated decay for specific mRNAs (e.g., DNA repair genes like OGG1), decoupling steady-state mRNA from protein accumulation. Hernández-Elvira, M. Sunnerhagen, P.  Post-transcriptional regulation during stress. FEMS Yeast Res. 2022, 22(1):foac025. doi: 10.1093/femsyr/foac025.

These processes are prominent in mainly neuronal models, where Pb-induced ROS and Ca²⁺ dysregulation amplify post-transcriptional control, leading to attenuated protein responses during neurotoxicity. Morgan, R.K.  Tapaswi, A.,  Polemi, K.M.,  Tolrud,  E.C., Bakulski, K.M, Svoboda, L.K., Dolinoy, D.C, Colacino, J.A. Environmentally Relevant Lead Exposure Impacts Gene Expression in SH-SY5Y Cells Throughout Neuronal Differentiation. Toxicological Sci. 2025, 207(2):435-448. doi: 10.1093/toxsci/kfaf072.

Point 2 The authors mainly use ELISA for quantify mitochondrial proteins. Other methods should be provided as alternative ways, such as Western blotting, or quantitative proteinomics, particularly for the gene showing opposite regulation at protein and mRNA level.

Answer: Thank you very much for this remark. We fully agree that alternative methods, such as Western blotting or quantitative proteomics, would be desirable; however, unfortunately, due to the lack of material and financial resources, it is currently impossible.

Point 3 For figure 8, the level of pTBK1 alter dramatically in different brains. So "TBK1-independent IFN-I pathway" is not conclusively. more samples should be tests. Were these samples collected or processed differently to cause the big changes?

Answer: We would like to thank the Reviewer for this valuable comment. We fully agree with your assessment, and in accordance with your suggestion, we expanded our analysis by examining additional samples. The newly obtained data are consistent with our previous results, showing no detectable changes in pTBK1(Ser172) immunoreactivity across all analysed structures. As in the earlier dataset, the current analysis also reveals substantial inter-individual variability in pTBK1(Ser172) levels among the animals.

We would also like to emphasise that all samples—both in the original and the extended dataset—were collected and processed under identical and strictly controlled conditions. This is supported by the comparable TBK1 and GAPDH levels observed across all examined brains, indicating the reliability and uniformity of sample preparation. Taken together, these findings further support our conclusion that the observed effects are mediated through a TBK1-independent IFN-I pathway.

In Figure 1 below, we present the results for pTBK1(Ser172) along with the corresponding Western blot images for pTBK1(Ser172) and GAPDH as a loading control. Additionally, in Figure 2, we have included the results for all analysed samples.

Figure 1. Immunoreactivity of p-TBK1(Ser172) in control (CTRL) and Pb-affected rats (Pb) was monitored using Western blot analysis. Densitometric analysis of p-TBK1, phosphorylated at (Ser172) in the Hippocampus (A), Cortex (B), and in the Cerebellum (C) was shown. Results were normalised to GAPDH. Data represent mean values ± SEM from n = 6 distinct animals from three different litters.

Figure 2. Immunoreactivity of p-TBK1(Ser172) in control (CTRL) and Pb-affected rats (Pb) was monitored using Western blot analysis. Densitometric analysis of p-TBK1, phosphorylated at (Ser172) in the Hippocampus (A), Cortex (B), and in the Cerebellum (C) was shown. Results were normalised to GAPDH. Data represent mean values ± SEM from all n = 12 distinct animals from six different litters.

Point 4 The resolution of Figure 1 should be increase, or provide zoom-in images to show mitochondria.

Answer: We would like to thank the Reviewer for this valuable comment. We fully agree with your opinion, so we provided the figure in Zoom to better visualise mitochondria.

Below, we provide an updated Figure 1B that includes enlarged images to improve the visualisation of mitochondria.

Figure 1B. Representative transmission electron microscopy images of neural mitochondria in control (A’, C’, E’) and Pb-exposed (B’, D’, F’) rats in three different areas of the brain: cerebellum (A’, B’), forebrain cortex (C’, D’) and hippocampus (E’, F’). (A’) Control cerebellar neuron with normal mitochondria – blue arrows, and likely dividing mitochondria – green arrow. (B’) Pb-exposed cerebellar neuron with altered mitochondria, highly elongated, fragmented, or deformed – red arrows. (C) Control forebrain cortical neuron with normal mitochondria – blue arrows. (D’) Pb-exposed forebrain cortical neuron with elongated, fragmented, or morphologically altered mitochondria – red arrows. (E’) Control hippocampal neuron with normal mitochondria – blue arrows. (F’) Pb-exposed hippocampal neuron with fragmented and morphologically abnormal mitochondria – red arrows.

Point 5 Table1 should be provided as a supplemental file.

Answer: Thank you very much for this remark. According to your request, Table 1 has been provided as Supplementary Materials.

Point 6 The y axis of qPCR graph should be presented as scientific natation way instead with decimal notation (such the scale of 0.000001)

Answer: Thank you very much for this remark. According to your request, the y-axis of the qPCR graph has been corrected and presented a scientific notation.

Below, we provide an updated Figure 6, where the y-axis of the qPCR graph for Mx1 has been corrected and presented a scientific notation.

Round 2

Reviewer 2 Report

Comments and Suggestions for Authors

The authors answered my questions and I'm satisfied with the revision.